

# Lidar observations of pyrocumulonimbus smoke plumes in the UTLS over Tomsk (Western Siberia, Russia) from 2000 to 2017

Vladimir V. Zuev[1], Vladislav V. Gerasimov[1,2], Aleksei V. Nevzorov[3], and Ekaterina S. Savelieva[1]

[1]Institute of Monitoring of Climatic and Ecological Systems SB RAS, Tomsk, 634055, Russia
[2]Tomsk State University, Tomsk, 634050, Russia
[3]V.E. Zuev Institute of Atmospheric Optics SB RAS, Tomsk, 634055, Russia

*Correspondence to*: Vladislav V. Gerasimov (gvvsnake@mail.ru)

**Abstract.** Large volcanic eruptions with the volcanic explosivity index (VEI) $\geq$ 3 are widely known to be the strongest source of long-lived aerosol in the upper troposphere and lower stratosphere (UTLS). However, the latest studies have revealed that massive forest (bush) fires represent another strong source of short-term (but intense) aerosol perturbations in the UTLS if combustion products from the fires reach these altitudes via convective ascent within pyrocumulonimbus clouds (pyroCbs). PyroCbs, generated by boreal wildfires in North America and North-East Asia and injecting smoke plumes into the UTLS, have been intensively studied using both ground- and space-based instruments since the beginning of the 21 century. In this paper, we focus on aerosol layers observed in the UTLS over Tomsk (56.48° N, 85.05° E, Western Siberia, Russia) that could be smoke plumes from such pyroCb events occurred in the 2000–2017 period. Using the HYSPLIT trajectory analysis, we have reliably assigned ten aerosol layers to nine out of more than 100 documented pyroCb events, the aftereffects of which could potentially be detected in the UTLS over Tomsk. All of the nine pyroCb events occurred in the USA and Canada: one event per year was in 2000, 2002, 2003, 2015, and 2016, whereas two events per year were in 2013 and 2017. No plumes from pyroCbs originating in the boreal zone of Siberia and the Far East (to the east of Tomsk) were observed in the UTLS over Tomsk between 2000 and 2017. We conclude that the lifetimes of pyroCb plumes to be detected in the UTLS using ground-based lidars are less than about a month, i.e. plumes from pyroCbs generated by wildfires to the east of Tomsk can significantly diffuse before reaching the Tomsk lidar station by the westerly zonal transport of air masses. A comparative analysis of the contributions from pyroCb events and volcanic eruptions with VEI $\geq$ 3 to aerosol loading of the UTLS over Tomsk has also been made. Finally, an aerosol plume from the Aleutian volcano Bogoslof erupted with VEI = 3 on 28 May 2017 was detected at altitudes between 10.8 and 13.5 km over Tomsk on 16 June 2017.

## 1 Introduction

There are many sources of aerosol in the troposphere: bio- and fossil-fuel burning, forest and bush fires, power generation and industrial processes, engines, volcanic eruptions, *etc*., and conversely, only a few such sources exist in the stratosphere. Aircraft emissions (combustion products of carbon-containing fuels) (Blake and Kato, 1995; Hendricks et al., 2004; Koehler et al., 2009; Wilkerson et al., 2010; Balkanski et al., 2010) and troposphere-to-stratosphere transport of air (Kremser et al.,





2016) are responsible for background aerosol loading in the lower stratosphere (LS). Large volcanic eruptions with the volcanic explosivity index (VEI) $\geq$ 3 represent the principal source of strong and long-term stratospheric aerosol perturbations (Robock, 2000; Robock and Oppenheimer, 2003; Kremser et al., 2016), which is confirmed by both space-borne and ground-based long-term lidar measurements (Vernier et al., 2011; Trickl et al., 2013; Mills et al., 2016; Sakai et

al., 2016; Khaykin et al, 2017; Zuev at al., 2017; Friberg et al., 2018). Volcanic plumes persist in the stratosphere for several months to several years, depending on the eruption latitude, VEI, and maximum plume altitude (MPA) after the eruptions (Hofmann et al., 2009). However, studies over the last two decades have revealed that, in addition to volcanic eruptions, there exists another source being able to cause short-term, but locally intense, aerosol perturbations in the LS. This source is massive forest (or bush) fires if combustion products from the fires reach stratospheric altitudes.

Massive forest fires (wildfires), the plumes of which can ascend to the LS, and their aftereffects have been intensively studied since the beginning of the 21 century (Fromm et al., 2000, 2005, 2006, 2008a,b, 2010; Fromm and Servranckx, 2003; Jost et al., 2004; Livesey et al., 2004; Damoah et al., 2006; Cammas et al., 2009; Gonzi and Palmer, 2010; Guan et al., 2010; Siddaway and Petelina, 2011; Dahlkötter et al., 2014; Paugam et al., 2016). Smoke plumes of the overwhelming majority of forest fires are located within the planetary boundary layer (Val Martin et al., 2010; Nikonovas et al., 2017; Rémy et al.,

2017), and a small number of them (< 5–10 %) can enter the free troposphere (Sofiev et al., 2013; Peterson et al., 2014). Only in exceptional cases aerosol plumes from the fires are able to reach stratospheric altitudes via convective ascent within pyro-cumulonimbus clouds (pyroCb; http://glossary.ametsoc.org/wiki/Pyrocumulonimbus). PyroCbs, injecting aerosol directly into the LS, originate mainly from boreal wildfires in North America (particularly in the Canadian boreal zone) and North-East Asia (Siberia and the Far East) (Fromm et al., 2010; Guan et al., 2010), and bush fires in Australia (Fromm et al.,

2006; Siddaway and Petelina, 2011). In particular years, pyroCb events can occur too frequently to be considered as an occasional source of aerosol in the LS. For example, Fromm et al. (2010) identified 17 such pyroCbs in the United States and Canada during the summer of 2002, a part of which reached the LS.

PyroCb stratospheric plumes can spread throughout the hemisphere and are detected by both ground- and space-based lidars for 2 to 4 months after their occurrence (Fromm et al., 2000, 2008b, 2010). Owing to their potential impact on the

climate, a lot of attention is currently paid to monitoring pyroCbs via, e.g., the Geostationary Operational Environmental Satellite (GOES) system (https://www.nasa.gov/content/goes). The data on pyroCb events occurring throughout the world are accumulated on the web page of the Cooperative Institute for Meteorological Satellite Studies (CIMSS): http://pyrocb.ssec.wisc.edu/ and their archives have been available since May 2013.

Ground-based lidar observations of stratospheric aerosol perturbations have been almost continuously performed in

Tomsk (56.48° N, 85.05° E, Western Siberia, Russia) for more than 30 years (Zuev et al., 1998, 2001, 2017). In the papers, we mainly discussed and focused on aerosol perturbations in the stratosphere over Tomsk after major volcanic eruptions (with VEI $\geq$ 3), the plumes of which were able to directly enter the stratosphere. To consider the effect of only volcanic eruptions on stratospheric aerosol loading and definitely exclude from consideration any aerosol perturbations in the upper troposphere (UT ) (such as cirrus clouds) and tropopause region, we analyzed the results of lidar measurements at altitudes




higher than 13–15 km. It is clear that this altitude limitation could lead to the loss of information on aerosol events like pyroCb plumes in the UTLS over Tomsk.

The possibility to observe stratospheric smoke plumes in Tomsk from massive forest fires occurred in North America was noted in Zuev et al. (2017). In this paper, we analyze all aerosol perturbations in the 11–30 km altitude region over

Tomsk that could be caused by massive wildfires in North America and North-East Asia from 2000 to 2017.

## 2 Lidar instruments and methods

The lidar measurements we consider were made using the aerosol channel of the Siberian Lidar Station (SLS) located in Tomsk. The transmitter of the channel represents a Nd:YAG laser (LS-2132T-LBO model, LOTIS TII Co., the Republic of Belarus) that operates at a wavelength of 532 nm with 100 mJ pulse energy and at a pulse repetition rate of 20 Hz. The

channel receiver is a Newtonian telescope with a mirror diameter of 0.3 m and a focal length of 1 m. The backscattered signals are registered by a photomultiplier tube R7206-01 (Hamamatsu Photonics, Japan) operating in the photon counting mode with a vertical resolution of 100 m. Owing to the rearrangement and improvement of the SLS, there were two shutdown periods of the aerosol channel from July 1997 to May 1999 and from February to September 2014. A detailed description of the SLS aerosol channel technical parameters is given in (Zuev, 2000; Burlakov et al., 2010).

The scattering ratio $R(H, \lambda)$ is used to describe the aerosol vertical distribution in the UTLS

$$R(H,\lambda) = \frac{\beta_\pi^m(H,\lambda) + \beta_\pi^a(H,\lambda)}{\beta_\pi^m(H,\lambda)} = 1 + \frac{\beta_\pi^a(H,\lambda)}{\beta_\pi^m(H,\lambda)}. \tag{1}$$

Here $\beta_\pi^m(H,\lambda)$ and $\beta_\pi^a(H,\lambda)$ are the altitude- and wavelength-dependent molecular (Rayleigh) and aerosol (Mie) backscatter coefficients, respectively; $\pi$ denotes the angle of the backscatter lidar signal propagation (i.e., $\pi$ radian). Stratospheric altitudes of ~30–35 km over Tomsk are mostly aerosol-free and, therefore, we use an altitude $H_0 = 30$ km for

calibrating the detected lidar signals by normalizing them to the molecular backscatter signal from $H_0$ (Zuev et al., 2017).

The integrated aerosol backscatter coefficient $B_\pi^a(\lambda)$ is used to discover the temporal dynamics of aerosol loading in the UTLS over Tomsk in the 2001–2017 period

$$B_\pi^a(\lambda) = \int_{H_1}^{H_2} \beta_\pi^a(H,\lambda)dH, \tag{2}$$

where the lower limit $H_1 = 11$ km falls within either the UT or LS due to the variability of the local tropopause altitude and

does not allow missing pyroCb plumes in the UTLS, and the upper limit is the calibration altitude $H_2 = H_0 = 30$ km.

When analyzing the perturbed scattering ratio $R(H, \lambda)$ profiles, cirrus clouds are excluded from consideration based on the following two criteria. First, a detected aerosol layer is definitely located in the UT and, second, the layer has a thickness of < 1 km and the value of $R(H) > 2.45$ for $\lambda = 532$ nm (see Appendix A). In some cases, however, there is a problem in





determining the location of detected aerosol layers (i.e., whether the layers are in the UT or LS) due to the absence of meteorological stations launching radiosondes in Tomsk. For this reason, to estimate the tropopause altitude over the lidar site, we use data for vertical temperature profiles from the three nearest to Tomsk meteorological stations launching radiosondes twice a day (at 00:00 and 12:00 UTC). These stations are located in Kolpashevo (58.31° N, 82.95° E),

Emeljanovo (56.18° N, 92.61° E), and Novosibirsk (54.96° N, 82.95° E) (Fig. 1), the radiosonde data of which can be found on the web page http://weather.uwyo.edu/upperair/sounding.html?region=np of the University of Wyoming (Kolpashevo, Emeljanovo, and Novosibirsk station numbers are 29231, 29572, and 29634, respectively).

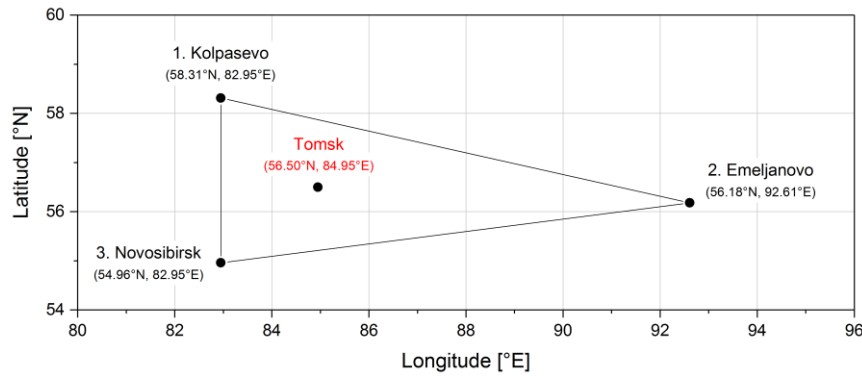

**Figure 1.** Three nearest to Tomsk meteorological stations launching radiosondes twice a day. The stations are numbered for convenience.

We took data on pyroCb events from scientific papers if the events were documented from 2000 to 2012, and at http://pyrocb.ssec.wisc.edu/ for pyroCbs occurred after May 2013. The required data on volcanic eruptions for the 2004–2017 period were taken from the Smithsonian Institution Global Volcanism Program (GVP; http://volcano.si.edu/; Section: Reports; Subsections: Smithsonian/USGS Weekly Volcanic Activity Report and Bulletin of the Global Volcanism Network). To assign aerosol layers detected in the UTLS over Tomsk to their sources (pyroCbs or volcanic eruptions), we analyze air

mass backward trajectories calculated with the NOAA's PC Windows-based HYSPLIT trajectory model (February 2018 Release; Stein et al., 2015; http://ready.arl.noaa.gov/HYSPLIT.php) and the HYSPLIT-compatible NOAA meteorological data from the NCEP/NCAR Reanalysis (2000 to 2003), the Global Data Assimilation System (GDAS) one-degree (July 2013 and May to August 2017) and half-degree (September 2013 to May 2016) archives. All altitudes $H$ of aerosol layers detected with the SLS aerosol channel, tropopause altitudes determined at the nearest meteorological stations, and MPAs

$H_{\text{MPA}}$ for pyroCbs determined with space-based instruments are given above sea level (a.s.l.), whereas altitudes $H_{\text{traj.}}^{\text{back.}}$ for the HYSPLIT air mass backward trajectories are calculated above ground level (a.g.l.). Since the SLS is situated at an altitude of 148 m a.s.l., the difference between altitudes $H$ (a.s.l.) and $H_{\text{traj.}}^{\text{back.}}$ (a.g.l.) for each initial point of the HYSPLIT backward trajectories (in the UTLS over Tomsk) is simply determined as $H - H_{\text{traj.}}^{\text{back.}} = 148$ m. All dates and times in this study are given in UTC.



## 3 Results of lidar observations in Tomsk for the 2000–2017 period

There were no eruptions of both tropical and northern volcanoes to be recorded at the SLS in Tomsk from the middle of 2000 to the end of 2004 and in the 2012–2016 period (Zuev et al., 2017), with the exception of the 13 February 2014 Kelut eruption, the plume of which, however, could not be detected at the SLS due to the 2014 shutdown period (see Sect. 2).

When analyzing aerosol layers observed over Tomsk and pyroCb events documented in the Northern Hemisphere over the period 2000–2017, we have discovered more than 100 pyroCbs (with known and unknown MPAs), the plumes of which could potentially be detected in the UTLS over Tomsk. However, only a few of the detected layers have been reliably attributed to the selected pyroCb events using the HYSPLIT trajectory analysis. To illustrate the correlation between the pyroCbs and corresponding layers over Tomsk, we present only the most successful examples of the HYSPLIT trajectories

that passed over or close to the places of origin of the pyroCbs (or near the known pyroCb plume locations when the exact coordinates and time of the pyroCb events are unknown).

### 3.1 Detection of pyroCb smoke plumes in the UTLS

The first aerosol layer we consider was observed in the UTLS over Tomsk at altitudes between 11.4 and 12.5 km with the maximum $R(H) = 2.42$ at $H = 12.1$ km a.s.l. on 4 September 2000 (Fig. 2a). The HYSPLIT trajectory analysis showed that

this layer was with high probability a smoke plume initially detected in the UTLS over Iowa (~42° N, ~92° W; USA) by the Total Ozone Monitoring Spectrometer (TOMS) on 27 August (Fromm et al., 2010). The plume originated from a pyroCb that occurred due to the massive "Jasper Fire" in the Black Hills National Forest (South Dakota, USA). Figure 2b shows, as an example, three air mass backward trajectories started from altitudes of 12.05–12.15 km a.s.l. over Tomsk at 19:30 UTC on 4 September and passed close to the Jasper Fire pyroCb plume location at altitudes $H_{\text{traj.}}^{\text{back.}}$ of 9.3–10.5 km a.g.l. on 27

August. Based upon the end points of the trajectories (with $H_{\text{traj.}}^{\text{back.}} \leq 10.5$ km a.g.l.) that are below latitude 45° N (Fig. 2b), the MPA $H_{\text{MPA}}$ did not exceed the tropopause at the place of the pyroCb origin.

    According to Fromm et al. (2010), a pyroCb generated by the "Mustang Fire" was registered with $H_{\text{MPA}} = 13$ km a.s.l. on the border of Utah and Wyoming (41.0° N, 109.3° W; USA) on 1 July 2002. Two weeks after the event, on 15 July, a comparatively "weak" aerosol layer was observed in the LS over Tomsk at altitudes of ~11.7–13.5 km with the maximum

$R(H) = 1.41$ at $H = 12.4$ km a.s.l. (Fig. 3a). Figure 3b presents the HYSPLIT air mass backward ensemble trajectories started from altitudes of ~12.1 km a.s.l. over Tomsk at 17:00 UTC on 15 July and passed near the place of origin of the Mustang Fire pyroCb at altitudes $H_{\text{traj.}}^{\text{back.}} \approx 12.5$–14.0 km a.g.l. on 2 July. The example trajectories allow us to assume that air masses containing the pyroCb plume were also spreading in the LS during the period from 2 to 15 July 2002.



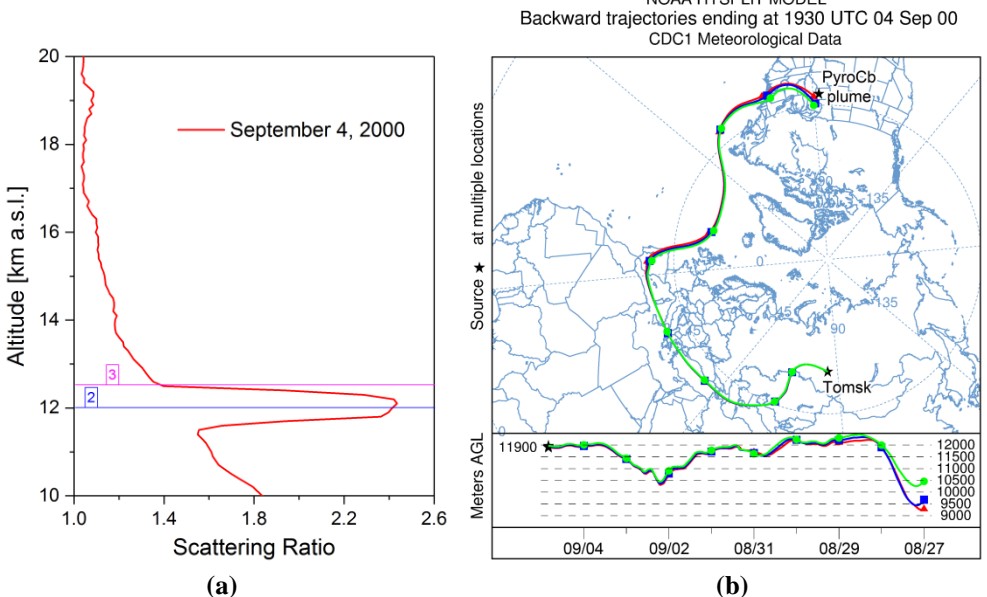

**Figure 2. (a)** Detection of the Jasper Fire pyroCb plume in the UTLS over Tomsk. The numbers 2 and 3 indicate the tropopause altitudes determined in Emeljanovo and Novosibirsk, respectively. **(b)** Air mass backward ensemble trajectories started from altitude of 12.05–12.15 km a.s.l. (11.9–12.0 km a.g.l.) over Tomsk at 19:30 UTC on 4 September 2000.

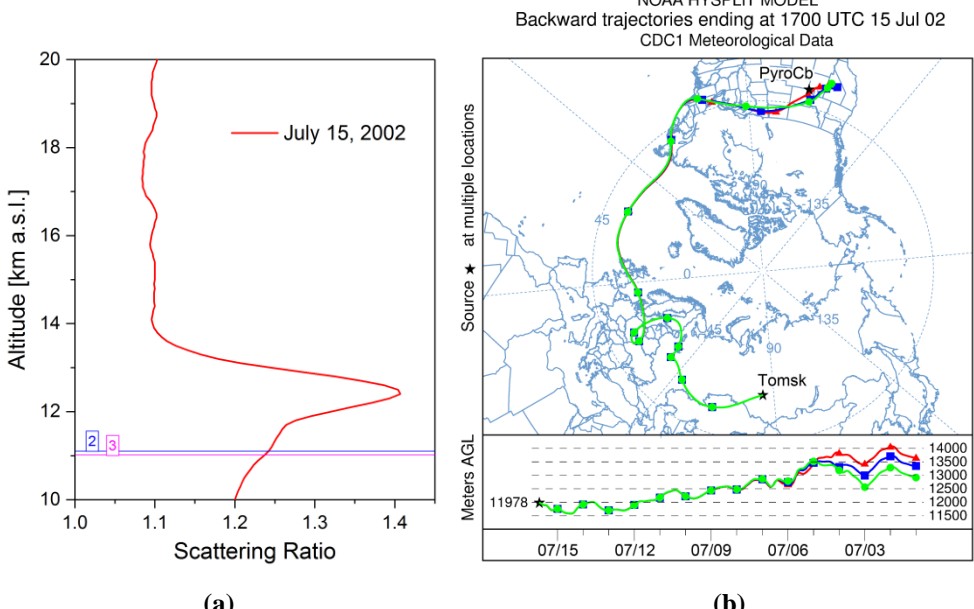

**Figure 3. (a)** Detection of the Mustang Fire pyroCb plume in the LS over Tomsk. The numbers 2 and 3 indicate the same as in Fig. 2a. **(b)** Air mass backward ensemble trajectories started from altitude of ~12.1 km a.s.l. over Tomsk at 17:00 UTC on 15 July 2002.



Another aerosol layer potentially associated with a pyroCb event was observed over Tomsk between 10 and 12 km with the maximum $R(H) = 1.87$ at $H = 11.1$ km a.s.l. on 29 August 2003 (Fig. 4a). Eleven days earlier, on 18 August, a pyroCb plume was registered in the UTLS over Hudson Bay (61° N, 89° W; Canada) by the TOMS. The pyroCb was previously generated by the "Conibear Lake Fire" in the Wood Buffalo National Park (Alberta/Northwest Territories, Canada) (Fromm et al., 2010). As seen in Fig. 4b, the HYSPLIT air mass backward trajectories, started from altitudes of ~11.75 km a.s.l. over Tomsk at 17:00 UTC on 29 August, passed over the pyroCb plume location at altitudes $H_{\text{traj.}}^{\text{back.}} \approx 11.7–12.0$ km a.g.l. on 18 August. Based on the behavior of the example trajectories (Fig. 4b) and the tropopause altitudes determined at the three nearest to Tomsk meteorological stations (Fig. 4a), we suppose that the pyroCb plume was spreading in the UT in a given period of time.

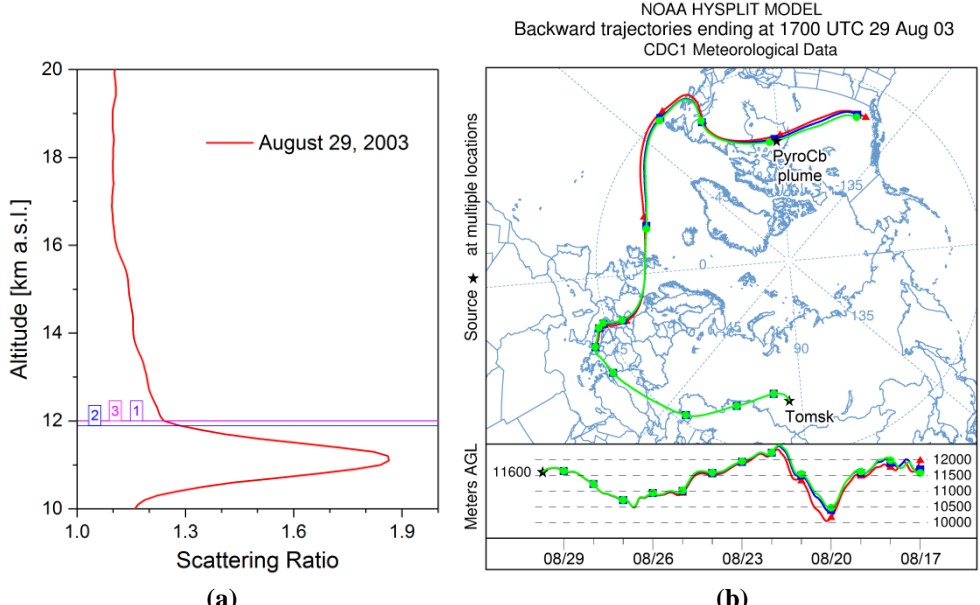

**Figure 4. (a)** Detection of the Conibear Lake Fire pyroCb plume in the UT over Tomsk. The numbers 1, 2, and 3 indicate the tropopause altitudes determined in Kolpashevo, Emeljanovo, and Novosibirsk, respectively. **(b)** Air mass backward ensemble trajectories started from altitude of ~11.75 km a.s.l. over Tomsk at 17:00 UTC on 29 August 2003.

The next two aerosol layers reliably attributed to pyroCb events were registered at the SLS in Tomsk only 10 years later, in July and September 2013. Namely, the first "weak" layer with the maximum $R(H) = 1.27$ at $H = 11.7$ km a.s.l. was observed in the UTLS over Tomsk on 14 July 2013 (Fig. 5a). The HYSPLIT trajectory analysis showed that the layer could represent a smoke plume from a pyroCb generated by large fires that were burning in the Eastmain region of Quebec (~52° N, ~78° W; Canada) in June–July 2013. The Eastmain pyroCb was discovered using the 1-km resolution GOES-13 0.63 $\mu$m visible channel after 21:55 UTC on 4 July (http://pyrocb.ssec.wisc.edu/archives/136). Figure 5b shows three example



HYSPLIT air mass backward trajectories started from altitudes of 10.6–10.7 km a.s.l. over Tomsk at 17:30 UTC on 14 July and passed near the place of origin of the Eastmain pyroCb at altitudes $H_{\text{traj.}}^{\text{back.}}$ of 5.5–9.0 km a.g.l. on 4 July. Based on the behavior of the trajectories, we suppose that the pyroCb plume was spreading in the troposphere during the period of time under consideration. Note that the aftereffects of the summer 2013 fire season in North America were also widely observed

5   in the troposphere over central Europe (Trickl et al., 2015; Markowicz et al., 2016).

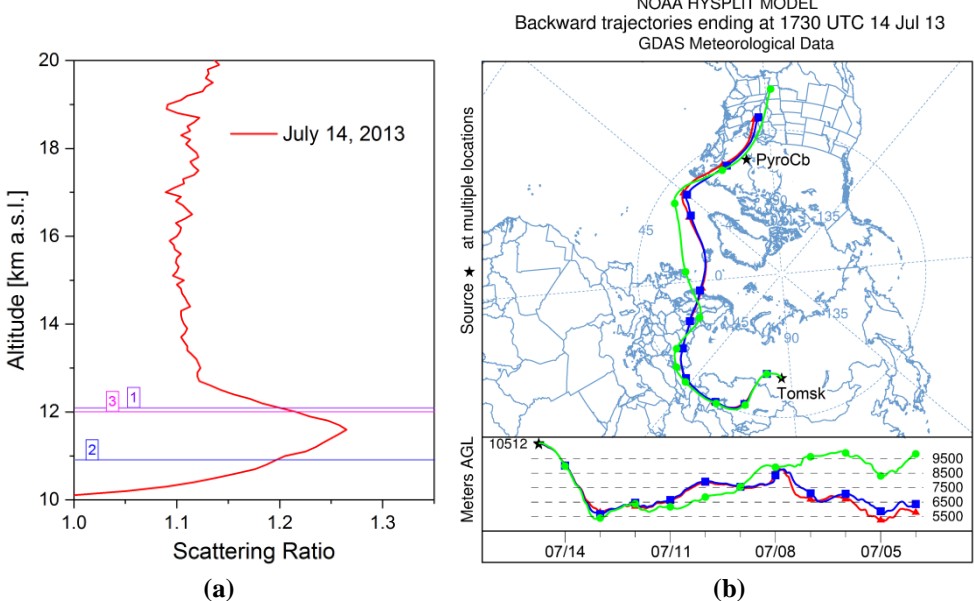

**(a)**                                                                                    **(b)**

**Figure 5. (a)** Detection of the Eastmain pyroCb plume in the UTLS over Tomsk. The numbers 1, 2, and 3 indicate the same as in Fig. 4a. **(b)** Air mass backward ensemble trajectories started from altitude of 10.6–10.7 km a.s.l. over Tomsk at 17:30 UTC on 14 July 2013.

10      The second "strong" layer was observed over Tomsk at altitudes between 11.2 and 12.8 km with the maximum $R(H) =$ 11.4 at $H = 11.8$ km a.s.l. on 23 September 2013 (Fig. 6a). A trajectory analysis showed that the layer can be assigned to a pyroCb event observed in British Columbia (~54° N, ~126° W; Canada) using the GOES-15 visible, shortwave IR, and longwave IR imageries between 23:30 UTC on 15 September and 02:30 UTC on 16 September (http://pyrocb.ssec.wisc.edu/archives/272). Three example HYSPLIT air mass backward trajectories that started from

15   altitudes of ~12.15 km a.s.l. over Tomsk at 17:30 UTC on 23 September and then passed close to the place of the pyroCb origin at altitudes $H_{\text{traj.}}^{\text{back.}} \approx 10.7$–11.7 km a.g.l. on 16 September are shown in Fig. 6b. Despite the high value of the scattering ratio $R(H)$, which is representative of cirrus clouds, the tropopause altitudes determined at the nearest to Tomsk meteorological stations show that the aerosol layer maximum was definitely in the LS (Fig. 6a). This allows us to conclude that the layer was a stratospheric one and could not be a cirrus cloud.



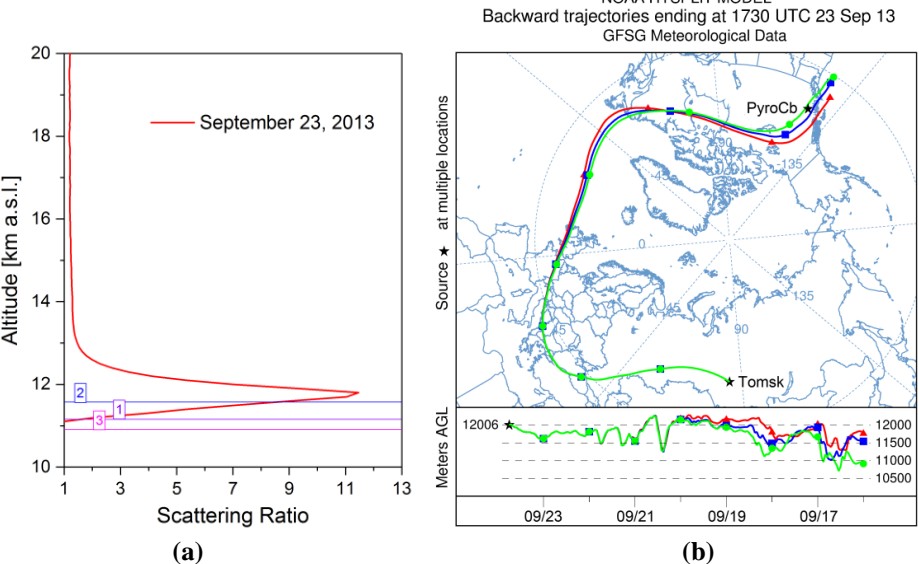

**Figure 6. (a)** Detection of a pyroCb plume from British Columbia in the LS over Tomsk. The numbers 1−3 indicate the same as in Fig. 4a.
**(b)** Air mass backward ensemble trajectories started from altitude of ~12.15 km a.s.l. over Tomsk at 17:30 UTC on 23 September 2013.

Only one aerosol layer associated with pyroCb events was detected in the LS over Tomsk in 2015. More precisely, the layer was observed between 10.1 and 12.0 km with the maximum $R(H) = 1.62$ at $H = 11.0$ km a.s.l. on 16 July (Fig. 7a). Five days before, on 11 July, two pyroCbs were registered in British Columbia using the GOES-15 0.63 $\mu$m visible, 3.9 $\mu$m IR, and 10.7 $\mu$m IR channels (http://pyrocb.ssec.wisc.edu/archives/985). The first pyroCb was observed at (56.4° N, 123.9° W) with $H_{MPA} = 10.5$ km a.s.l. around 00:30 UTC, whereas the second one was detected at (52.2° N, 124° W) with $H_{MPA} = 10$ km a.s.l. 4.5 hours later, at ~05:00 UTC. As seen in Fig. 7b, three HYSPLIT backward trajectories that started from altitudes of 10.3–10.5 km a.s.l. over Tomsk at 18:00 UTC on 16 July passed near and over the place of the former pyroCb origin at altitudes $H_{\text{traj.}}^{\text{back.}} \approx 10.8$–12.0 km a.g.l. on 11 July. Based on the tropopause altitudes determined at the nearest meteorological stations (Fig. 7a) and the behavior of the example trajectories (Fig. 7b), we can assume that the pyroCb plume was spreading in the LS in a given period of time (11–16 July 2015). We could not connect the aerosol layer under consideration with the latter pyroCb event.

Another marked pyroCb formed in British Columbia was observed at (~56° N, ~122° W) by the GOES-15 visible and IR channels at 22:00 UTC on 16 May 2016 (http://pyrocb.ssec.wisc.edu/archives/1622). Eleven days after the event, on 27 May, a thin aerosol layer with a thickness of ~0.8 km and the maximum $R(H) = 2.48$ at $H = 11.3$ km a.s.l. was detected in the UTLS over Tomsk (Fig. 8a). Figure 8b presents the HYSPLIT backward ensemble trajectories started from altitudes of ~11.75 km a.s.l. over Tomsk at 17:30 UTC on 27 May and then passed close to the place of the pyroCb origin at altitudes $H_{\text{traj.}}^{\text{back.}} \approx 10.5$–11.3 km a.g.l. on 16 May. As seen in Fig. 8a, it is difficult to definitely determine whether the aerosol layer was in the UT or LS over Tomsk. Nevertheless, the fact that the layer was completely higher than 11 km and two out of three




tropopause altitudes allows us to conclude that the layer was not a cirrus cloud. The smoke from the pyroCb was also observed in the UTLS over the UK with Raman lidars between 23 and 31 May 2016. (Vaughan et al., 2018).

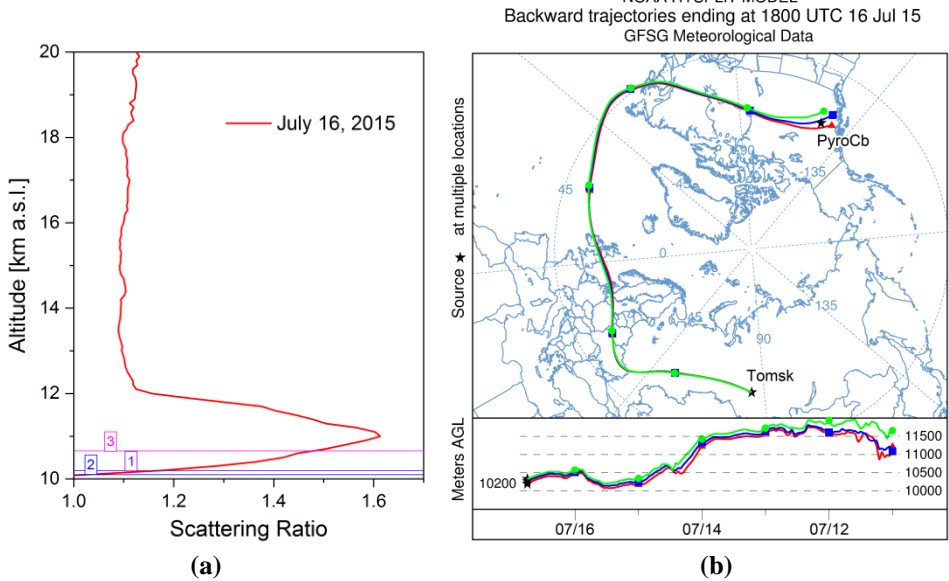

(a)  (b)

5  **Figure 7. (a)** Detection of a pyroCb plume from British Columbia in the LS over Tomsk. The numbers 1–3 indicate the same as in Fig. 4a.
**(b)** Air mass backward ensemble trajectories started from altitude of 10.3–10.5 km a.s.l. over Tomsk at 18:00 UTC on 16 July 2015.

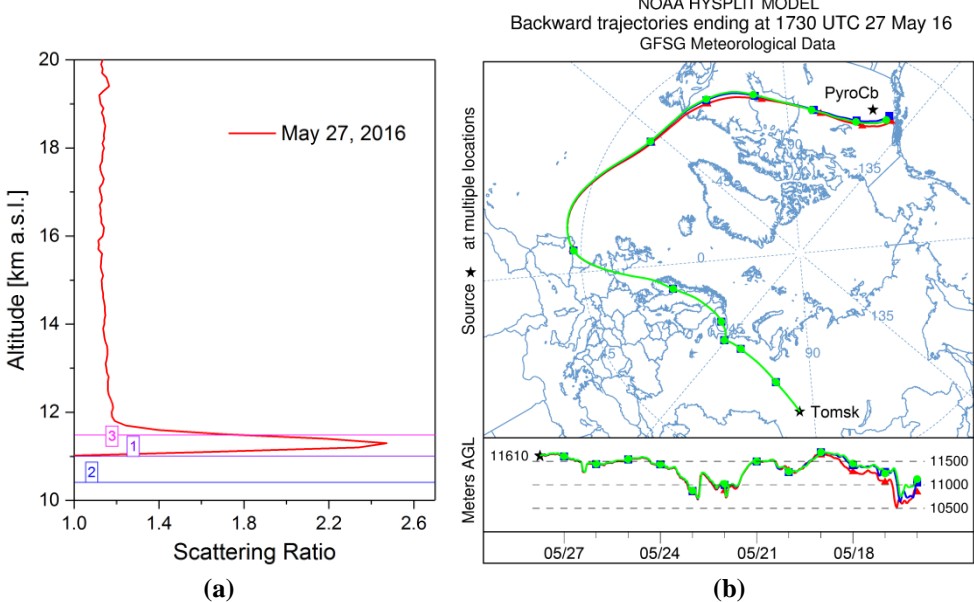

(a)  (b)

**Figure 8. (a)** Detection of a pyroCb plume from British Columbia in the UTLS over Tomsk. The numbers 1–3 indicate the same as in Fig.
10  4a. **(b)** Air mass backward ensemble trajectories started from altitude of ~11.75 km a.s.l. over Tomsk at 17:30 UTC on 27 May 2016.



In August 2017, massive forest fires in British Columbia generated several strong pyroCbs, the plumes of which reached stratospheric altitudes. The aftereffects of these Canadian wildfires and pyroCb events were widely observed in the UTLS over Europe in August and September 2017 and have already been intensively studied by different research groups (Ansmann et al., 2018; Haarig et al., 2018; Khaykin et al., 2018; Hu et al., 2018). We could attribute three aerosol layers observed in the LS over Tomsk at the end of August to two out of five pyroCbs detected by the GOES-15 instruments at (51.8° N, 123.2° W) and (53.1° N, 121.0° W) around 03:30 and 05:30 UTC on 12 August, respectively (http://pyrocb.ssec.wisc.edu/archives/2135). The first ("strong" and thin) aerosol layer was observed between 15.0 and 15.8 km with the maximum $R(H) = 5.8$ at $H = 15.4$ km a.s.l. two weeks after the event, on 26 August (Fig. 9a). Three days later, on 29 August, the second ("weak") layer was detected with the maximum $R(H) = 1.37$ at $H = 14.5$ km a.s.l. (Fig. 9b). Finally, the third layer was observed between 14.3 and 16.2 km with the maximum $R(H) = 3.1$ at $H = 15.7$ km a.s.l. on 31 August (Fig. 9c). In each case considered, the HYSPLIT trajectory analysis showed that the backward ensemble trajectories passed near and/or over the places of origin of both pyroCbs on 12 August (Fig. 10). The initial conditions (times and altitudes over Tomsk) for each trajectory can also be found in Fig. 10. Based upon the end points of the trajectories, the MPAs $H_{MPA}$ for both pyroCbs were definitely in the LS within the range of 13.5–15.0 km a.g.l. We cannot exclude that the layers observed over Tomsk on 26, 29, and 31 August could contain aerosol from the other three pyroCbs detected by the NOAA-18 instruments on the evening of 12 August.

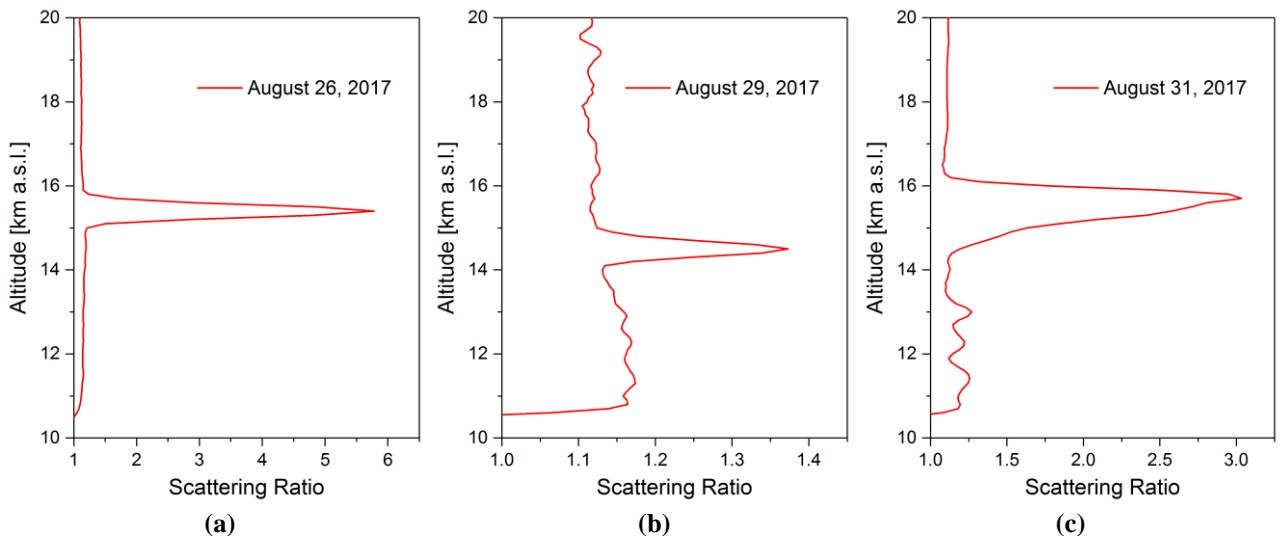

**Figure 9. (a), (b), (c)** Detection of pyroCb plumes from British Columbia in the LS over Tomsk on 26, 29, and 31 August 2017, respectively.





(a)

(b)

(c)

**Figure 10.** Air mass backward ensemble trajectories started from altitude of **(a)** ~15.75 km a.s.l. over Tomsk at 17:00 UTC on 26 August 2017, **(b)** ~14.55 km a.s.l. at 16:00 UTC on 29 August 2017, and **(c)** ~14.9 km a.s.l. at 15:00 UTC on 31 August 2017.

## 3.2 Detection of the Bogoslof volcanic plume in 2017

The period between 2012 and 2016 was, in general, a period of volcanic quiet, during which no significant volcanic eruptions (with VEI ≥ 3) occurred in the Northern Hemisphere, with the exception of the 13 February 2014 Kelut eruption (GVP). However, there was a series of short-term explosive eruptions with VEI ≤ 3 of the Aleutian volcano Bogoslof from



the end of December 2016 to August 2017 (GVP). A noteworthy aerosol layer was detected in the UTLS over Tomsk at altitudes between 10.8 and 13.5 km with the maximum $R(H) = 1.31$ at $H = 11.7$ km a.s.l. on 16 June 2017 (Fig. 11a). The aerosol layer was initially considered as a smoke plume from a pyroCb that was generated by a wildfire started to burn at (58.3° N, 106.2° E; Russia) on 25 May 2017 (http://pyrocb.ssec.wisc.edu/archives/2021), but our attempts to assign the layer

to the pyroCb event have failed. However, the HYSPLIT trajectory analysis showed that this layer was with high probability a plume from Bogoslof volcano erupted with $H_{MPA} = 13.7$ km a.s.l. at 23:16 UTC on 28 May 2017 (GVP). Figure 11b shows three example backward trajectories started from altitudes of 12.7–12.8 km a.s.l. over Tomsk at 18:00 UTC on 16 June and passed close to the Bogoslof volcano location at altitudes $H_{traj.}^{back.} \approx 12.0$–12.7 km a.g.l. on 28–29 May. Hence, at least the upper part of the aerosol layer located in the LS over Tomsk can be considered as an aftereffect of the 28 May 2017

Bogoslof volcano eruption.

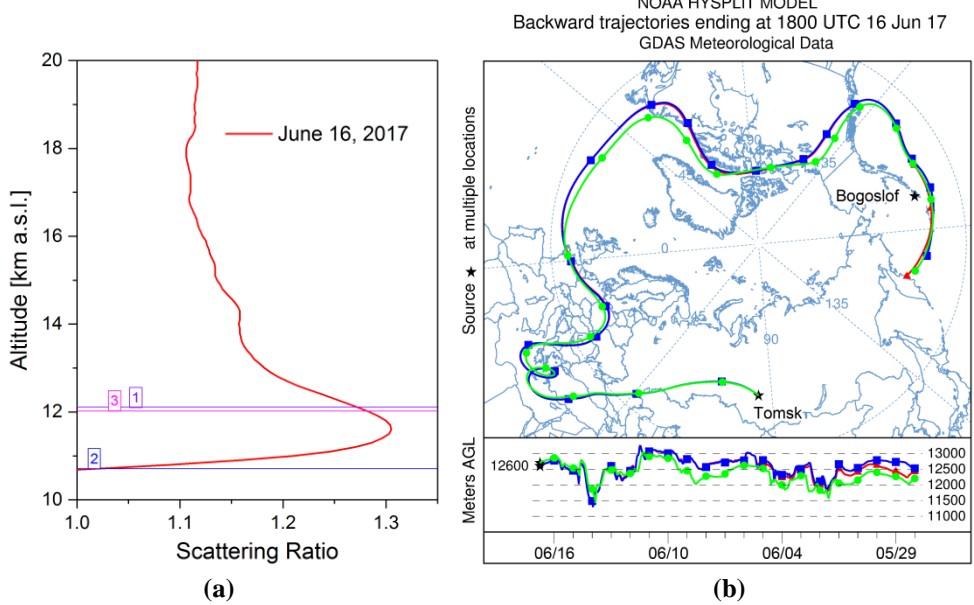

**Figure 11. (a)** Detection of the Bogoslof volcanic plume in the UTLS over Tomsk. The numbers 1–3 indicate the same as in Fig. 4a. **(b)** Air mass backward trajectories started from altitude of 12.7–12.8 km a.s.l. over Tomsk at 18:00 UTC on 16 June 2017.

**3.3 PyroCb events in 2004–2012**

Several strong pyroCbs, the plumes of which reached UTLS altitudes with $H_{MPA} \geq 12$ km a.s.l. and could potentially be detected in the UTLS over Tomsk, were documented in the Northern Hemisphere between 2004 and 2011 (Table 1). However, no aerosol layers associated with these pyroCb events were observed at the SLS during the period. This was due to unfavorable weather conditions or pyroCb plumes could have diffused or passed by the SLS and, therefore, might not be

detected. Note also that twelve explosive eruptions with VEI = 3–4 of both tropical and northern extratropical volcanoes, the



aftereffects of which were reliably registered in the stratosphere over Tomsk, occurred in the 2004–2011 period (Zuev at al., 2017). We do not exclude that pyroCb plumes could hardly be discerned against the background of the volcanic plumes in the UTLS over Tomsk in this period. There were no significant events (volcanic eruptions and pyroCbs) to be recorded at the SLS in 2012.

**Table 1.** List of some documented pyroCbs with $H_{\mathrm{MPA}} \geq 12$ km a.s.l. occurred in the Northern Hemisphere, the plumes of which could potentially be detected in the UTLS over Tomsk, in the 2004–2011 period.

| Plume date | PyroCb plume location | $H_{\mathrm{MPA}}$, km | Reference |
|---|---|---|---|
| 30 Jun 2004 | 43.1° N, 89.4° W | 13.0 | Damoah et al., 2006 |
| 27 Jul 2006 | 64.5° N, 114.5° E | 12.6 | Guan et al., 2010; Gonzi and Palmer, 2010 |
| 5 Sep 2006 | 48.5° N, 89.5° W | 12.3 | Guan et al., 2010 |
| 10 Jun 2007 | 39.5° N, 122.5° E | 15.6 | Guan et al., 2010 |
| 8 Jul 2007 | 33.5° N, 104.5° W | 12.0 | Guan et al., 2010 |
| 27 Jul 2008 | 60.7° N, 114.4° W | 12.0 | Paugam et al., 2016 |
| 12 Sep 2011 | 47.9° N, 91.5° W | 13.6 | Dahlkötter et al., 2014 |

## 4 Time series of the integrated aerosol backscatter coefficient (2001–2017)

To estimate the contribution of the pyroCb events discussed above to aerosol loading of the UTLS over Tomsk, we have analyzed the 2001–2017 time series of the aerosol backscatter coefficient $B_{\pi}^{\mathrm{a}}$ values, obtained from the SLS observations at

$\lambda = 532$ nm and integrated over the 11–30 km altitude range. The upper part of Fig. 12 presents the 10-day average $B_{\pi}^{\mathrm{a}}$ values with the annual average $B_{\pi}^{\mathrm{a}}$ ones assigned to 1 July of each year. PyroCb events and volcanic eruptions (Tables 2 and 3), the plumes of which were observed in the UTLS over Tomsk between 2000 and 2017, are indicated by red and black vertical bars, respectively, in the lower part of Fig. 12.

The time series can be divided into three intervals with different trends in the annual average $B_{\pi}^{\mathrm{a}}$ values: 2001–2004,

2004–2011, and 2011–2017 (Fig. 12). The first 2001–2004 interval (a) is marked by a negative trend in the annual average $B_{\pi}^{\mathrm{a}}$ values caused by the absence of volcanic eruptions with VEI ≥ 3 in the Northern Hemisphere. Despite the fact that three pyroCb plumes were detected from 2000 to 2003, the minimum (background) annual average $B_{\pi}^{\mathrm{a}} = 3.07 \times 10^{-4}$ sr$^{-1}$ was reached in 2004. Note that when integrating the aerosol backscatter coefficient $\beta_{\pi}^{\mathrm{a}}(H, \lambda)$ over the 15–30 km altitude range, the minimum annual average $B_{\pi}^{\mathrm{a}} = 1.29 \times 10^{-4}$ sr$^{-1}$ is also reached in 2004 (Zuev et al., 2017). During the second 2004–2011

interval (b), the aftereffects of twelve volcanic eruptions (Table 3) measurably perturbed the UTLS over Tomsk and,





therefore, were registered at the SLS. This volcanic activity led to a positive trend in the annual average $B_\pi^a$ values. The last 2011–2017 interval (c) is characterized by comparatively low activity of both tropical and northern volcanoes. Namely, only two volcanic eruptions that could perturb the UTLS over Tomsk occurred for a given period of time (Table 3). Six pyroCb events injected smoke into the UTLS in 2013 and 2015–2017 (Table 2) resulted, however, in a negative trend in the annual

average $B_\pi^a$ values.

The trends in Fig. 12 definitely show that for Tomsk region the aftereffects of tropical and northern volcanic eruptions with VEI ≥ 3 are stronger and longer-lasting than those of pyroCb events occurred mainly due to wildfires in North America. Indeed, volumes and lifetimes of primary (volcanic ejecta) and secondary (sulfuric acid) aerosols in the UTLS from explosive volcanic eruptions are known to be higher (Hofmann et al., 2009) compared to those of aerosols from pyroCb

plumes (Fromm et al., 2010). Hence, twelve volcanic eruptions occurred in time interval (b) naturally led to an increase in aerosol loading of the UTLS over Tomsk and, therefore, to a positive trend in the annual average $B_\pi^a$ values. PyroCbs generated by wildfires from 2004 to 2011 (including documented ones listed in Table 1) also had to perturb the UTLS over Tomsk, but we could not unambiguously discern the pyroCb plumes against the background of more powerful volcanic plumes observed during this period. Therefore, the positive trend in the period 2004 to 2011 should have been mostly caused

by volcanic eruptions (the same conclusion was reached by Zuev et al. (2017), when integrating $\beta_\pi^a(H, \lambda)$ over the 15–30 km altitude range). The presence of pyroCb plumes in the UTLS over Tomsk in periods (a) and (c) did not allow the annual average $B_\pi^a$ values to decrease to the background level (as it was in 2004 in the absence of both volcanic and pyroCb plumes). On the other hand, plumes from two or more pyroCbs that have occurred in North America in a single year are able to markedly increase aerosol loading of the UTLS over Tomsk compared to the previous year (Fig. 12). For example, the

annual average $B_\pi^a$ reached a value of $5.34 \times 10^{-4}$ $sr^{-1}$ due to pyroCbs 4 and 5 (Table 2) occurred in 2013 (by comparison, $B_\pi^a = 4.20 \times 10^{-4}$ $sr^{-1}$ in 2012) and $B_\pi^a = 4.34 \times 10^{-4}$ $sr^{-1}$ due to pyroCbs 8 and 9 together with the Bogoslof eruption occurred in 2017 ( $B_\pi^a = 3.78 \times 10^{-4}$ $sr^{-1}$ in 2016). This substantiates the assumption that the effect of pyroCbs on aerosol loading of the UTLS sometimes can be comparable to that of volcanic eruptions (Fromm et al., 2010).



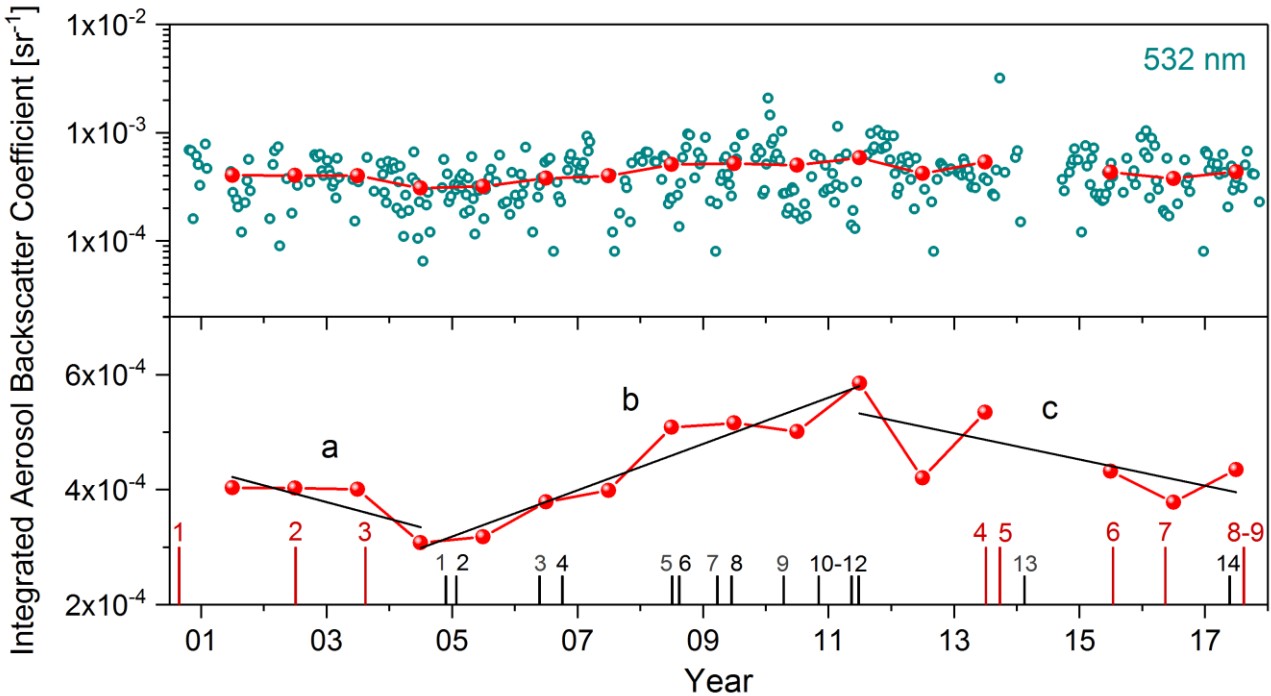

**Figure 12.** The 2001–2017 time series of the integrated aerosol backscatter coefficient $B_\pi^a$ obtained from lidar observations at λ = 532 nm in the 11–30 km altitude range. Open dark-green circles denote the 10-day average $B_\pi^a$ values; solid red circles show the annual average $B_\pi^a$ values assigned to 1 July of each year. Red and black vertical bars in the lower part of the figure indicate, respectively, pyroCbs and volcanic eruptions (see also Tables 2 and 3), the plumes of which were observed in the UTLS over Tomsk between 2000 and 2017.

**Table 2.** List of documented pyroCbs, the plumes of which perturbed the UTLS over Tomsk during the 2000–2017 period.

| N | Plume date | PyroCb event location | PyroCb plume location | $H_{MPA}$, km |
|---|---|---|---|---|
| 1 | 27 Aug. 2000 [a] | | 42° N, 92° W | |
| 2 | 1 July 2002 [a] | 41.0° N, 109.3° W | | 13 |
| 3 | 18 Aug. 2003 [a] | | 61° N, 89° W | |
| 4 | 4 July 2013 [b] | 52° N, 78° W | | |
| 5 | 16 Sept. 2013 [b] | 54° N, 126° W | | |
| 6 | 11 July 2015 [b] | 56.4° N, 123.9° W | | 10.5 |
| 7 | 16 May 2016 [b] | 56° N, 122° W | | |
| 8 | 12 Aug. 2017 [b] | 51.8° N, 123.2° W | | |
| 9 | 12 Aug. 2017 [b] | 53.1° N, 121° W | | |

[a] Fromm et al., 2010

[b] pyrocb.ssec.wisc.edu



**Table 3.** List of volcanic eruptions that have perturbed the UTLS over Tomsk from 2004 to the present day. The list was retrieved from the GVP data.

| N | Date/Period | Volcano | Location | $H_{MPA}$, km | VEI |
|---|---|---|---|---|---|
| 1 | 24 Nov. 2004 | Manam | Papua New Guinea (4.1° S, 145.0° E) | 18 | 4 |
| 2 | 27 Jan. 2005 | Manam | Papua New Guinea (4.1° S, 145.0° E) | 24 | 4 |
| 3 | 20 May 2006 | Soufriere Hills | West Indies (16.7° N, 62.2° W) | 17 | 4 |
| 4 | 7 Oct. 2006 | Rabaul | Papua New Guinea (4.3° S, 152.2° E) | 18 | 4 |
| 5 | 12 Jul. 2008 | Okmok | Aleutian Islands (53.4° N, 168.1° W) | 15 | 4 |
| 6 | 7 Aug. 2008 | Kasatochi | Aleutian Islands (52.2° N, 175.5° W) | 14 | 4 |
| 7 | 22 Mar. 2009 | Redoubt | Alaska (60. 5° N, 152.7° W) | 20 | 3 |
| 8 | 11–16 Jun. 2009 | Sarychev Peak | Kuril Islands (48.1° N, 153.2° E) | 21 | 4 |
| 9 | 14–17 Apr. 2010 | Eyjafjallajökull | Iceland (63.6° N, 19.6° W) | 9 | 4 |
| 10 | 4–5 Nov. 2010 | Merapi | Indonesia (7.5° S, 110.4° E) | 18.3 | 4 |
| 11 | 21 May 2011 | Grimsvötn | Iceland (64.4° N, 17.3° W) | 20 | 4 |
| 12 | 13 Jun. 2011 | Nabro | Eritrea (13.4° N, 41.7° E) | 13.7* | 4 |
| 13 | 13 Feb. 2014 | Kelut | Indonesia (7.9° S, 112.3° W) | 17 | 4 |
| 14 | 28 May 2017 | Bogoslof | Aleutian Islands (53.9° N, 168.0° W) | 13.7 | 3 |

* ~18 km (Fromm et al., 2014)

## 5 Concluding remarks

The increasing number and intensity of boreal forest fires in North America and North-East Asia due to climate warming for the last decades (Wotton et al., 2010; Sofiev et al., 2013; Rémy et al., 2017) lead to an increasing number of pyroCbs, the plumes of which are able to reach the UTLS (Fromm et al., 2010; Guan et al., 2010). Boreal wildfires are usually active during the warm half year (April to September) and spread in the UTLS for long distances mainly due to the westerly zonal transport of air masses in the Northern Hemisphere. Therefore, the plumes of pyroCbs occurred in North America are frequently detected in the UTLS over Europe, and more rarely over Siberia, and the Far East by both ground- and space-based lidars.

In this study, we have considered and analyzed aerosol layers in the UTLS (11–30 km) over Tomsk that could represent smoke plumes from pyroCbs generated by massive wildfires in North America and North-East Asia between 2000 and 2017. Using the HYSPLIT trajectory analysis, we have reliably assigned ten such layers to nine out of more than 100 documented pyroCb events, the aftereffects of which could potentially be detected at the SLS. All of the nine pyroCb events occurred in North America: one event per year was in 2000, 2002, 2003, 2015, and 2016, whereas two events per year were in 2013 and 2017. Such a small number of observed pyroCbs could be due to unfavorable weather conditions or pyroCb plumes could



have passed by the SLS. Unfortunately, we could not unambiguously discern plumes from pyroCbs occurred in the 2004–2011 period (Table 1) against the background of more powerful plumes from twelve volcanic eruptions observed during this period (Table 3).

Massive forest fires generating pyroCbs are also known to occur in North-East Asia (pyrocb.ssec.wisc.edu). However, no plumes in the UTLS over Tomsk from pyroCbs occurred in the boreal zone of Siberia and the Far East (to the east of Tomsk) were detected at the SLS between 2000 and 2017. We can assume that the lifetimes of pyroCb plumes to be detected in the UTLS using ground-based lidars are less than about a month. In other words, plumes from pyroCbs generated by wildfires to the east of Tomsk can significantly diffuse before reaching the SLS due to the westerly zonal transport. This probably explains a comparatively "low" contribution from pyroCbs to aerosol loading of the UTLS over Tomsk and, therefore, the negative trends in the annual average $B_\pi^a$ values in the absence of, and at a low, volcanic activity in time intervals (a) and (c), respectively (Fig. 12).

Based on the results of lidar observations at the SLS between 2000 and 2017, we can conclude the following. During a short-term period (up to three weeks) after pyroCb events have occurred in North America, their aftereffects in the UTLS over Tomsk are comparable to those of volcanic eruptions with VEI ~ 3. The 10-day average $B_\pi^a$ value after the events can be even higher than that after volcanic eruptions. For example, the 10-day average $B_\pi^a$ for the 20–30 September period reached the maximum value of $3.20 \times 10^{-3}$ sr$^{-1}$ after pyroCb event 5 (Table 2) of 16 September 2013 (Fig. 12). Moreover, smoke plumes reached the UTLS over Tomsk from two or more pyroCbs in a single year can lead to a marked increase in aerosol loading compared to that in the previous year. For example, the annual average $B_\pi^a$ value increased by 27.1% in 2013 and 14.8% in 2017 (together with the 2017 Bogoslof eruption). Nevertheless, the contribution from pyroCbs (generated by wildfires in North America and injecting smoke into the UTLS) to the annual average $B_\pi^a$ value integrated over the 11–30 km altitude range is noticeably lower (for Tomsk region) than the contribution from both tropical and northern volcanic eruptions with VEI ≥ 3 (due to, among other things, secondary sulfuric acid aerosol).

## 6 Data availability

The NOAA's HYSPLIT model used to calculate all air mass backward trajectories is available at http://ready.arl.noaa.gov/HYSPLIT.php. The volcanic eruption data we used can be found at http://volcano.si.edu/ and the data on pyroCb events occurred after May 2013 are located at http://pyrocb.ssec.wisc.edu/. The radiosonde data for the Kolpashevo, Emeljanovo, and Novosibirsk meteorological stations are on the web page http://weather.uwyo.edu/.



*Author contributions.* VVZ and VVG performed main analysis of all data and wrote the paper. AVN made measurements at the SLS and processed lidar data. ESS performed the HYSPLIT trajectory analysis. VVG and ESS retrieved data on pyroCbs, the plumes of which could potentially be detected in the UTLS over Tomsk.

*Competing interests.* The authors declare that they have no conflict of interest.

*Acknowledgements.* We thank Michael Fromm (the US Naval Research Laboratory) for the information on the aerosol cloud coming to Tomsk from several strong pyroCbs generated by wildfires in British Columbia (Canada) in August 2017.

**Appendix A: Scattering ratio $R(H, \lambda)$ values for cirrus clouds**

Aerosol layers detected in the UT with ground-based lidars are identified as cirrus clouds if the scattering ratio $R(H) > 10$ for a laser wavelength $\lambda_1 = 532$ nm (Tao et al., 2008; Samokhvalov et al., 2013). However, according to Sassen et al. (1989), the

minimum value of $R(H)$ can be 5.2 in the case of invisible to the naked eye co-called "subvisual" cirrus clouds (for a laser wavelength $\lambda_2 = 694.3$ nm) with a thickness of $< 1$ km. To calculate the minimum $R(H)$ value for $\lambda_1 = 532$ nm, one can use the fact that the aerosol backscatter coefficient $\beta_\pi^a(H, \lambda)$ is considered to be independent of the scattered light wavelength if aerosol particles are much greater than the wavelength (Measures, 1984). Since cirrus cloud particles ($\sim 25\ \mu$m, Sassen et al., 1989) are greater than both considered wavelengths $\lambda_1$ and $\lambda_2$, we can assume $\beta_\pi^a(\lambda_1) = \beta_\pi^a(\lambda_2)$ for each altitude $H$.

Therefore, using Eq. (1), we can write the following equality

$$[R(\lambda_1) - 1]\, \beta_\pi^m(\lambda_1) = [R(\lambda_2) - 1]\, \beta_\pi^m(\lambda_2). \tag{A1}$$

Taking into account the dependence $\beta_\pi^m(\lambda) \sim \lambda^{-4}$, for the scattering ratio we have

$$R(\lambda_1) = 1 + [R(\lambda_2) - 1]\left(\lambda_1/\lambda_2\right)^4. \tag{A2}$$

Substituting $\lambda_1 = 532$ nm, $\lambda_2 = 694.3$ nm, and $R(\lambda_2) = 5.2$ into Eq. (A2), we finally obtain $R(\lambda_1) = 2.45$.

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
