# Peer review of "Lidar observations of pyrocumulonimbus smoke plumes in the UTLS over Tomsk (Western Siberia, Russia) from 2000 to 2017"

_Atmospheric Chemistry and Physics, 2018_

## Referee Comment (RC1) · Anonymous Referee #2 · 17 Dec 2018

The contribution is an important documentation of pyrocumulonimbus-related aerosol events occurring in the upper troposphere and lower stratosphere. Long-term lidar observations taken over Tomsk, Siberia, Russia, from 2000-2017 are carefully analysed, presented, and discussed. The result is a well written paper that even may guide other lidar groups to re-analyze their own lidar observations. Nevertheless, only few stations around the world can provide such results as presented here.

I recommend: Minor revisions.

Details

Abstract: Remove the first lines... Start with: In this paper... The abstract should be

always compact and as short as possible: Goals, methods, key results, not more. All motivating points shall be given in the Introduction (only).

P2, L13-15: Satellite remote sensing is not able to provide us with the top of the smoke layers, the retrieved tops are at much too low altitudes. That should be clearly mentioned. Only lidars are able to resolve smoke plumes correctly. Satellites often provide the erroneous impression that most of the smoke is in the PBL, which is contradiction with almost all lidar observations around the world (e.g., as monitored by EARLINET teams of Amiridis et al., ACP and JGR, Nicolae et al., JGR, and also Mattis et al., JGR 2008. . .). So, on a global scale, only CALIOP can do a reliable job.

P3, L21-25: Any comment? Why not using always the tropopause as H1 in Eq.(2)? The tropopause height is always available from GDAS. . . And the reader (at least this reviewer) wants to obtain a clear picture of the smoke impact on stratospheric aerosol conditions.

P7, P15: No event from 2003 to 2013? Can you say something about the reasons? Was it wet in western Canada, western United states? Or was the long range transport blocked?

P15, L6-24: What about bad weather conditions? ..and the probability that you missed several short-term PyroCB events. . .? Is your lidar automated? Probably not, so the probability is at least not zero that you missed some nice events.

———————————————————

---

## Referee Comment (RC2) · Fromm (Referee) · 20 Dec 2018

Reviewer: Mike Fromm

Note: Throughout this review I will refer to the author team as "auth."

This paper is a broad survey of a single aerosol lidar data set covering 18 years. The instrument's data record has been examined by this team in prior papers, focused solely on the lower stratosphere (LS) and volcanic aerosols. In this work auth nudge their reportable lower data bound down to 11 km, i.e. into the upper troposphere (UT). Their aim is to expand their scope from volcanic plumes to include pyrocumulonimbus

smoke plumes.

Auth are to be commended for their rigorous accounting for pyroCb events. They use published works and a pyroCb blog to identify a large number of pyroCb events, from which plumes might have crossed over the Tomsk lidar. Considering that the Tomsk aerosol lidar is positioned in a region otherwise poorly instrumented for aerosol profiling, and has been operated for years before space-based lidar data became available, this is a strategic data set. And considering that auth have methodically undertaken an accounting for UTLS pyroCb smoke, this stands as a first to my knowledge. Hence this paper merits consideration in ACP.

Auth make many convincing connections between Tomsk UTLS aerosol layers and specific pyroCb events. However, there are also a few unconvincing cases reported here. Auth also attempt to attribute a weak UT aerosol layer with Bogoslof volcano over a very long trajectory path. I found this to be unconvincing. It may be possible to bolster each one of the less than convincing cases but substantial work is needed to do so. For instance, during the CALIPSO era, the space-based lidar data can be used to corroborate the Tomsk observations and infer particle type (based on depolarization). Auth cite an example of a work (Vaughan et al.) that performed such an analysis. Perhaps auth might follow that example in testing their connections.

My recommendation is for auth to make the substantial changes needed to make all the cases convincing or remove those that are not improvable.

One general concern is the use of both "above ground level" and "above mean sea level" altitude reference frames. I strongly recommend that auth use just ASL. AGL can be confusing in the HYSPLIT plots because altitude variations of a UTLS air parcel often have nothing to do with the ground, yet the AGL plots make it look like big excursions are occurring when in many cases it's just because of topographic changes. HYSPLIT does allow one to plot the time series in the ASL reference frame, so this valuable improvement would come with little effort.

It is a good idea to consider UT aerosol instead of the high LS cutoff used in prior papers. Auth now have chosen a fixed altitude (11 km) that is sometimes in the UT and sometimes in the LS. They need to defend the choice of this fixed altitude.

Another general concern is that auth regularly refer to "weak" and "strong" aerosol layers but they do not define the terms. I would suggest that if they want to continue using those qualifiers, to establish a quantifiable distinction up front.

It is surprising that auth do not find any UTLS aerosol layers attributable to pyroCbs in the 9-year period 2004-2012, given their tabulation of strong pyroCb events and the fact that a number of convincing connections were made before and since. In addition to the table provided we know that the frequency of pyroCbs was roughly the same in this period as in others. In fact the pyroCb community has been able to discern pyroCb smoke and volcanic sulfates in the UTLS at the same time (e.g. in July 2011, CALIPSO showed Grimsvotn, Nabro sulfates and pyroCb smoke from the Las Conchas pyroCb (New Mexico) over Europe). It may be beyond the scope of this paper to revisit this period, but I would ask auth to provide an accounting of the dates of lidar measurements deemed to be cirrus free. Such a table in an appendix or supporting information section would be of great value to researchers assessing the lidar data coverage through the years.

The manuscript, marked with comment bubbles containing minor and technical suggestions, is included with this review.

Please also note the supplement to this comment:
https://www.atmos-chem-phys-discuss.net/acp-2018-1153/acp-2018-1153-RC2-supplement.pdf
* * *
[Figure]

**Supplement:**

[revised manuscript text omitted]

---

## Author Comment (AC1) · 28 Feb 2019

**Manuscript Number:** acp-2018-1153
**Manuscript Type:** Research article
**Title:** Lidar observations of pyrocumulonimbus smoke plumes in the UTLS over Tomsk (Western Siberia, Russia) from 2000 to 2017

**Point-by-point response to Referee 2**

**General comment**

**Comment:** The contribution is an important documentation of pyrocumulonimbus-related aerosol events occurring in the upper troposphere and lower stratosphere. Long-term lidar observations taken over Tomsk, Siberia, Russia, from 2000-2017 are carefully analysed, presented, and discussed. The result is a well written paper that even may guide other lidar groups to re-analyze their own lidar observations. Nevertheless, only few stations around the world can provide such results as presented here.
I recommend: Minor revisions.

**Response:** We thank Referee 2 for a positive review and useful comments. Our point-by-point responses to Reviewer #1 comments and changes made are presented below.

**Specific comments**

**Comment 1:** Abstract: Remove the first lines… Start with: In this paper… The abstract should be always compact and as short as possible: Goals, methods, key results, not more. All motivating points shall be given in the Introduction (only).
**Response 1:** Perhaps, you are right, but we follow the manuscript preparation guidelines for authors provided by ACP. Namely: "The abstract should be intelligible to the general reader without reference to the text. After a brief introduction of the topic, the summary recapitulates the key points of the article and mentions possible directions for prospective research…"

https://www.atmospheric-chemistry-and-physics.net/for_authors/manuscript_preparation.html

**Comment 2:** P2, L13-15: Satellite remote sensing is not able to provide us with the top of the smoke layers, the retrieved tops are at much too low altitudes. That should be clearly mentioned. Only lidars are able to resolve smoke plumes correctly. Satellites often provide the erroneous impression that most of the smoke is in the PBL, which is contradiction with almost all lidar observations around the world (e.g., as monitored by EARLINET teams of Amiridis et al., ACP and JGR, Nicolae et al., JGR, and also Mattis et al., JGR 2008: : :). So, on a global scale, only CALIOP can do a reliable job.

**Response 2:** We agree and have never declared in our manuscript that satellite remote sensing provides us with the top of the smoke layers. Due to this reason, only 2 maximum pyroCb plume altitudes are known (see Table 2). We estimated the altitudes from which air masses containing pyroCb plumes arrived in Tomsk from North America based on the HYSPLIT backward trajectories. However, we now use available CALIPSO data (version 4.10) to corroborate the Tomsk observations for the 2013–2017 period. The CALIPSO data are given in the Supplement.

**Comment 3:** P3, L21-25: Any comment? Why not using always the tropopause as H1 in Eq.(2)? The tropopause height is always available from GDAS… And the reader (at least this reviewer) wants to obtain a clear picture of the smoke impact on stratospheric aerosol conditions.

**Response 3:** The use of the fixed 11-km altitude is a compulsory measure. As we noted in the manuscript, we are not able to precisely determine the tropopause altitude above the lidar site due to the absence of meteorological stations launching radiosondes in Tomsk. So, we use sonde data from the three nearest to Tomsk stations (launching sondes twice a day), which allow us to estimate the tropopause altitude more precisely (and closer in time to aerosol layer observation) than that from GDAS data. Due to an 11-km fixed altitude and,

therefore, a fixed 11–30 km altitude region, we can (regardless of the real tropopause altitude) make a comparative analysis of aerosol loading over Tomsk from both volcanic eruptions and pyroCb events in the 2000– 2017 period. In addition, a fixed altitude of 11 km excludes the tropospheric aerosol sources and does not allow us to miss pyroCb plumes from North America.

**Instead of**
"where the lower limit $H_1$ = 11 km falls within either the UT or LS due to the variability of the local tropopause altitude and does not allow missing pyroCb plumes in the UTLS, and the upper limit is the calibration altitude $H_2 = H_0 = 30$ km."

**We write**
"where the lower limit $H_1$ = 11 km can fall within the UT, TR* or LS due to the variability of the local tropopause altitude and the upper limit is the calibration altitude $H_2 = H_0 = 30$ km. The use of the fixed 11-km altitude is a compulsory measure because there is a problem in determining the tropopause altitude over the lidar site due to the absence of a meteorological station launching radiosondes in Tomsk. Nevertheless, the 11-km lower limit does not allow missing pyroCb plumes from Northern America and excludes the tropospheric aerosol sources with the exception of cirrus clouds. Moreover, the fixed 11–30 km altitude region allows us (regardless of the real tropopause altitude) to make a comparative analysis of aerosol loading over Tomsk due to both volcanic eruptions and pyroCb events from 2000 to 2017."
*TR means tropopause region
[**Page 4, lines 2–8, revised manuscript**]

**Comment 4:** P7, P15: No event from 2003 to 2013? Can you say something about the reasons? Was it wet in western Canada, western United states? Or was the long range transport blocked?

**Response 4:** Several pyroCb events, the plumes from which could potentially be detected in Tomsk in the 2004–2011 period, are listed in Table 1. Section 3 reports the facts of pyroCb plume detection in the UTLS over Tomsk, whereas the reasons of the absence of detected pyroCb aftereffects in Tomsk are discussed in Sections 4 and 5. According to our findings and the conclusions provided, e.g., by Peterson et al. (2018), pyroCb aftereffects are comparable to those from volcanic eruptions with VEI ≤ 3, whereas 11 out of 12 volcanic eruptions, detected in Tomsk in the 2004–2011 period, had VEI = 4. Therefore, the main reason is that we cannot unambiguously discern the pyroCb plumes against the background of more powerful volcanic plumes through the use of only one-wavelength aerosol ground-based lidar. The use of space-based lidar measurement data to infer particle type in the 2004–2011 volcanic period is not the subject of the current research. Nevertheless, we could draw some conclusions about the pyroCb smoke impact on UTLS aerosol conditions due to two periods of volcanic quiescence (2001–2004 and 2012–2017), during which no significant volcanic eruptions (with VEI ≥ 3) occurred in the Northern Hemisphere (see Sections 4 and 5).

Peterson, D.; Campbell, J.; Hyer, E.; Fromm, M.; Kablick, G.; Cossuth, J.; DeLand, M. Wildfire-driven thunderstorms cause a volcano-like stratospheric injection of smoke. NPJ Clim. Atmos. Sci. 2018, 1. https://www.nature.com/articles/s41612-018-0039-3

**Comment 5:** P15, L6-24: What about bad weather conditions? … and the probability that you missed several short-term PyroCB events…? Is your lidar automated? Probably not, so the probability is at least not zero that you missed some nice events.

**Response 5:** You are right. Bad (cloudy) weather conditions led to the absence of lidar data for 290 out of 630 (~46%) ten-day periods from 2000 to 2017. See please "Data for Figure 10.opj" in the Supplement. To open the file, the scientific graphing and data analysis software "Origin" is required (https://www.originlab.com/), the trial version of which can be downloaded at: https://www.originlab.com/demodownload.aspx. Taking into account the 2004–2011 volcanic period, we can say that many pyroCb events could be definitely missed.

Sincerely,
Authors

---

## Author Comment (AC2) · 28 Feb 2019

**Manuscript Number:** acp-2018-1153
**Manuscript Type:** Research article
**Title:** Lidar observations of pyrocumulonimbus smoke plumes in the UTLS over Tomsk (Western Siberia, Russia) from 2000 to 2017

**Point-by-point response to Dr. Fromm**

**General comments**

**Comment I:** This paper is a broad survey of a single aerosol lidar data set covering 18 years. The instrument's data record has been examined by this team in prior papers, focused solely on the lower stratosphere (LS) and volcanic aerosols. In this work auth nudge their reportable lower data bound down to 11 km, i.e. into the upper troposphere (UT). Their aim is to expand their scope from volcanic plumes to include pyrocumulonimbus smoke plumes.

Auth are to be commended for their rigorous accounting for pyroCb events. They use published works and a pyroCb blog to identify a large number of pyroCb events, from which plumes might have crossed over the Tomsk lidar. Considering that the Tomsk aerosol lidar is positioned in a region otherwise poorly instrumented for aerosol profiling, and has been operated for years before space-based lidar data became available, this is a strategic data set. And considering that auth have methodically undertaken an accounting for UTLS pyroCb smoke, this stands as a first to my knowledge. Hence this paper merits consideration in ACP.

**Response I:** We deeply thank Dr. Fromm for his interest in our work as well as for valuable comments, questions and suggestions which allowed us to improve our manuscript.

**Comment II:** Auth make many convincing connections between Tomsk UTLS aerosol layers and specific pyroCb events. However, there are also a few unconvincing cases reported here. Auth also attempt to attribute a weak UT aerosol layer with Bogoslof volcano over a very long trajectory path. I found this to be unconvincing. It may be possible to bolster each one of the less than convincing cases but substantial work is needed to do so. For instance, during the CALIPSO era, the space-based lidar data can be used to corroborate the Tomsk observations and infer particle type (based on depolarization). Auth cite an example of a work (Vaughan et al.) that performed such an analysis. Perhaps auth might follow that example in testing their connections.

My recommendation is for auth to make the substantial changes needed to make all the cases convincing or remove those that are not improvable.

**Response II:** In general, we agree with Dr. Fromm's standpoint and have made some corrections and changes in the manuscript. The main goal of our paper is to demonstrate the possibility to observe pyroCb plumes from North America in the UTLS over Tomsk using a one-wavelength aerosol ground-based lidar, and we showed several cases of pyroCb plume detection. All results and conclusions we presented are based on the Tomsk lidar measurements, HYSPLIT trajectory analysis, and data on pyroCb events from scientific papers if the events were documented from 2000 to 2012 and at http://pyrocb.ssec.wisc.edu/ for pyroCbs that occurred after May 2013. Nevertheless, we have analyzed available CALIPSO data (for the 2013 – 2017 period) to corroborate the Tomsk observations and, as a result, we have excluded two cases from consideration. First, the aerosol layer detected on 23 September 2013 with high probability represented a cirrus cloud and, second, we have found no strong evidence that the aerosol layer detected over Tomsk on 16 June 2017 was an aftereffect of the 28 May 2017 Bogoslof volcano eruption. The corresponding parts of the text have been removed from the revised manuscript. The CALIPSO data for the other cases of aerosol layers detected in the UTLS over Tomsk are given in the supplementary materials.

The Dr. Fromm's proposal to use the CALIPSO data to infer particle type via depolarization measurements in the 2004–2012 period is very interesting and valuable, but is the subject of future and extensive research.

**We have added the following sentences to the revised manuscript:**

"We have also analyzed available CALIPSO data to corroborate the Tomsk observations for the 2013–2017 period. The CALIPSO data are given in the Supplement."
[**Page 5, lines 17–18, revised manuscript**]

**Comment III:** One general concern is the use of both "above ground level" and "above mean sea level" altitude reference frames. I strongly recommend that auth use just ASL. AGL can be confusing in the HYSPLIT plots because altitude variations of a UTLS air parcel often have nothing to do with the ground, yet the AGL plots make it look like big excursions are occurring when in many cases it's just because of topographic changes. HYSPLIT does allow one to plot the time series in the ASL reference frame, so this valuable improvement would come with little effort.

**Response III:** We agree and have recalculated all the HYSPLIT air mass backward trajectories "above mean sea level" (AMSL). Thus, all altitudes are now given AMSL, whereas all abbreviations "a.s.l." and "a.g.l." have been deleted in the revised manuscript.

**Instead of**

~~"All altitudes $H$ of aerosol layers detected with the SLS aerosol channel, tropopause altitudes determined at the nearest meteorological stations, and MPAs $H_{MPA}$ for pyroCbs determined with space-based instruments are given above sea level (a.s.l.), whereas altitudes $H_{traj.}^{back.}$ for the HYSPLIT air mass backward trajectories are calculated above ground level (a.g.l.). Since the SLS is situated at an altitude of 148 m a.s.l., the difference between altitudes $H$ (a.s.l.) and $H_{traj.}^{back.}$ (a.g.l.) for each initial point of the HYSPLIT backward trajectories (in the UTLS over Tomsk) is simply determined as $H - H_{traj.}^{back.} = 148$ m.~~ All dates and times in this study are given in UTC."

**We write**

"All altitudes in this study are given above mean sea level (AMSL), whereas all dates and times are given in UTC."
[**Page 5, line 6, revised manuscript**]

**Comment IV:** It is a good idea to consider UT aerosol instead of the high LS cutoff used in prior papers. Auth now have chosen a fixed altitude (11 km) that is sometimes in the UT and sometimes in the LS. They need to defend the choice of this fixed altitude.

**Response IV:** Since there is a problem in determining the tropopause altitude over Tomsk due to the absence of meteorological stations launching radiosondes in Tomsk, we use a fixed altitude of 11 km that: 1) is clearly situated in the UTLS region, 2) does not allow us to miss pyroCb plumes from North America, and 3) excludes the tropospheric aerosol sources with the exception of cirrus clouds. The clouds are excluded from consideration based on the criteria presented in Appendix A. The fixed 11–30 km altitude region allows making a comparative analysis of aerosol loading over Tomsk due to both volcanic eruptions and pyroCb events from 2000 to 2017.

**Instead of**

"where the lower limit $H_1 = 11$ km falls within either the UT or LS due to the variability of the local tropopause altitude and does not allow missing pyroCb plumes in the UTLS, and the upper limit is the calibration altitude $H_2 = H_0 = 30$ km."

**We write**

"where the lower limit $H_1 = 11$ km can fall within the UT, TR* or LS due to the variability of the local tropopause altitude and the upper limit is the calibration altitude $H_2 = H_0 = 30$ km. The use of the fixed 11-km altitude is a compulsory measure because there is a problem in determining the tropopause altitude over the lidar site due to the absence of a meteorological station launching radiosondes in Tomsk. Nevertheless, the 11-km lower limit does not allow missing pyroCb plumes from Northern America and excludes the tropospheric aerosol sources with the exception of cirrus clouds. Moreover, the fixed 11–30 km altitude region allows us (regardless of the real tropopause altitude) to make a comparative analysis of aerosol loading over Tomsk due to both volcanic eruptions and pyroCb events from 2000 to 2017."
*TR means tropopause region
[**Page 4, lines 2–8, revised manuscript**]

**Comment V:** Another general concern is that auth regularly refer to "weak" and "strong" aerosol layers but they do not define the terms. I would suggest that if they want to continue using those qualifiers, to establish a quantifiable distinction up front.

**Response V:** The adjectives "weak" and "strong" applied for aerosol layers are only comparative ones. To avoid misunderstandings, we have removed them from the text.

**Comment VI:** It is surprising that auth do not find any UTLS aerosol layers attributable to pyroCbs in the 9-year period 2004-2012, given their tabulation of strong pyroCb events and the fact that a number of convincing connections were made before and since. In addition to the table provided we know that the frequency of

pyroCbs was roughly the same in this period as in others. In fact the pyroCb community has been able to discern pyroCb smoke and volcanic sulfates in the UTLS at the same time (e.g. in July 2011, CALIPSO showed Grimsvotn, Nabro sulfates and pyroCb smoke from the Las Conchas pyroCb (New Mexico) over Europe). It may be beyond the scope of this paper to revisit this period, but I would ask auth to provide an accounting of the dates of lidar measurements deemed to be cirrus free. Such a table in an appendix or supporting information section would be of great value to researchers assessing the lidar data coverage through the years.

**Response VI:** As was noted in **Response II**, the aerosol particle type determination in the period of volcanic activity (2004–2011) is not the subject of the current research. Our conclusion that aftereffects of pyroCb events are comparable to those of volcanic eruptions with VEI ~3 is in agreement with the findings obtained by Peterson et al. (2018), which clearly showed that even multiple pyroCb events cannot compete with volcanic eruptions with VEI = 4. Taking into account that 11 out of 12 volcanic eruptions in the 2004–2011 period had VEI = 4, we can conclude that these eruptions were the principal source of aerosol loading in the UTLS over Tomsk from 2004 to 2011 and plumes from individual pyroCb events could not be discerned against the background of the volcanic plumes using our lidar.

The information on cirrus cloud detection will not be of interest to pyroCb community because only three cirrus clouds at altitudes higher than the lower limit of integration $H_1$ = 11 km (see Eq. (2)) were detected in the warm half year (on 22 and 26 June 2012 and 23 September 2013). The other cases were observed in the cold (October–March) half year when no pyroCb events occurred in the boreal zone of North America and North-East Asia. These three "warm" cirrus clouds together with a couple of the cold-half-year cirrus clouds are presented below. The criteria to discern cirrus clouds have been corrected in Appendix A of the revised manuscript.

Peterson, D., Campbell, J., Hyer, E., Fromm, M., Kablick, G., Cossuth, J., and DeLand, M.: Wildfire-driven thunderstorms cause a volcano-like stratospheric injection of smoke, NPJ Clim. Atmos. Sci., 1, 30, https://doi.org/10.1038/s41612-018-0039-3, 2018.

[Figure]

The numbers 1, 2, and 3 indicate the tropopause altitudes estimated in Kolpashevo, Emeljanovo, and Novosibirsk, respectively, on the corresponding day.

**Specific comments**

(marked in the manuscript with comment bubbles containing minor and technical suggestions)

**Abstract**

**Comment 1: page 1, line 20**. Do you mean "age" rather than "lifetime?" Presumably these plumes didn't all reach Tomsk at the end of their detectable lifetime.

**Response 1:** To avoid misunderstandings, we substituted "lifetime of" with "time duration for" here and in Section 5.

**Instead of**
"We conclude that the lifetimes of pyroCb plumes to be detected in the UTLS…"
**We write**
"We conclude that the time durations for pyroCb plumes to be detected in the UTLS…"
[**Page 1, line 20, and Page 17, line 5, revised manuscript**]

**Comment 2: page 1, line 24.** Auth's findings should be summarized here in the abstract.

**Response 2:** We have added the conclusions drawn from a comparative analysis of the contributions from pyroCb events and volcanic eruptions with VEI ≥ 3 to aerosol loading of the UTLS over Tomsk to the abstract. The reference to the 28 May 2017 Bogoslof volcanic eruption was removed from the abstract (see also **Response II**).

**Instead of**
"A comparative analysis of the contributions from pyroCb events and volcanic eruptions with VEI ≥ 3 to aerosol loading of the UTLS over Tomsk has also been made. "
**We write**
"A comparative analysis of the contributions from pyroCb events and volcanic eruptions with VEI ≥ 3 to aerosol loading of the UTLS over Tomsk showed the following. Plumes from two or more pyroCbs that have occurred in North America in a single year are able to markedly increase the aerosol loading compared to the previous year. The annual average value of the integrated aerosol backscatter coefficient $B_\pi^{\mathrm{a}}$ increased by 14.8% in 2017 compared to that in 2016 due to multiple pyroCbs occurred in British Columbia (Canada) in August 2017. The aftereffects of pyroCb events are comparable to those of volcanic eruptions with VEI ≤ 3, but even multiple pyroCbs can hardly compete with volcanic eruptions with VEI = 4."
[**Page 1, lines 22–28, revised manuscript**]

**1 Introduction**

**Comment 3: page 3, line 3**. Consider combining this paragraph with the previous one. I think it would improve the line of argument.

**Response 3:** OK. Done.

**2 Lidar instruments and methods**

**Comment 4: page 3, line 16, Eq. (1).** Since this is a single wavelength lidar, this formula can be simplified. Consider using a "532" subscript and eliminating the lambda dimension.

**Response 4:** We agree with this suggestion and now use $R_{532}(H)$, $\beta_{\pi,532}^{\mathrm{m}}(H)$, $\beta_{\pi,532}^{\mathrm{a}}(H)$, and $B_{\pi,532}^{\mathrm{a}}$ instead of $R(H,\lambda)$, $\beta_\pi^{\mathrm{m}}(H,\lambda)$, $\beta_\pi^{\mathrm{a}}(H,\lambda)$, and $B_\pi^{\mathrm{a}}(\lambda)$, respectively, in Section 2. The shorter notation $R(H)$ is used in the other sections.

**Comment 5: page 3, line 16, Eq. (1).** Technical point. "z" is the more commonly accepted symbol for height.

**Response 5:** We would like to use "H" for altitude, because we used the same symbol in our previous paper (Zuev et al., 2017).

Zuev, V. V., Burlakov, V. D., Nevzorov, A. V., Pravdin, V. L., Savelieva, E. S., and Gerasimov, V. V.: 30-year lidar observations of the stratospheric aerosol layer state over Tomsk (Western Siberia, Russia), Atmos. Chem. Phys., 17, 3067-3081, https://doi.org/10.5194/acp-17-3067-2017, 2017.

**Comment 6: page 3, line 27.** Presumably auth are trying to articulate how they treat the UT part of the integrating column. This needs to be clarified.

**Response 6:** We determine the UT part via analyzing the tropopause altitudes estimated from radiosonde data of three neighbor meteorological stations. We have revised the atmospheric regions where cirrus clouds can be and slightly changed Appendix A.

**Instead of**
"When analyzing the perturbed scattering ratio $R(H, \lambda)$ profiles, cirrus clouds are excluded from consideration based on the following two criteria. First, a detected aerosol layer is definitely located in the UT and, second, the layer has a thickness of < 1 km and the value of $R(H) > 2.45$ for $\lambda = 532$ nm (see Appendix A)."
**We write**
"When analyzing perturbed scattering ratio $R_{532}(H)$ profiles, cirrus clouds are excluded from consideration based on the criteria presented in Appendix A."
[**Page 4, lines 16–17, revised manuscript**]

In Appendix A:

**Instead of**
"Aerosol layers detected in the UT with ground-based lidars are identified as cirrus clouds if the scattering ratio $R(H) > 10$ for a laser wavelength $\lambda_1 = 532$ nm (Tao et al., 2008; Samokhvalov et al., 2013). However, according to Sassen et al. (1989), the minimum value of $R(H)$ can be 5.2 in the case of invisible to the naked eye co-called "subvisual" cirrus clouds (for a laser wavelength $\lambda_2 = 694.3$ nm) with a thickness of < 1 km."
**We write**
"Aerosol layers detected in the UT/TR with ground-based lidars are identified as cirrus clouds if the scattering ratio $R(H) > 10$ for a laser wavelength $\lambda_1 = 532$ nm (Tao et al., 2008; Samokhvalov et al., 2013). However, according to Sassen et al. (1989), the minimum value of $R(H)$ can be 5.2 in the case of invisible to the naked eye co-called "subvisual" cirrus clouds (for a laser wavelength $\lambda_2 = 694.3$ nm) with a thickness of < 1 km. Note, however, that the thickness of other cirrus cloud types can often be more than 1 km (Goldfarb et al., 2001)."

Goldfarb, L., Keckhut P., Chanin, M.-L., and Hauchecorne, A.: Cirrus climatological results from lidar measurements at OHP (44° N, 6° E), Geophys. Res. Lett., 28, 1687–1690, https://doi.org/10.1029/2000GL012701, 2001.
[**Page 18, lines 8–12, revised manuscript**]

**Comment 7: page 3, line 28**. Delete this if auth adopt the "532" subscript.

**Response 7:** This comment is no longer relevant due to the changes made in a revised manuscript (see also the changes after **Response 6**).

**Comment 8: page 4, line 1**. Consider simplifying to "tropopause-relative location."

**Response 8:** This comment is no longer relevant due to the changes made in a revised manuscript (see also the changes after **Response IV**).

**Instead of**
"In some cases, however, there is a problem in determining the location of detected aerosol layers (i.e., whether the layers are in the UT or LS) due to the absence of meteorological stations launching radiosondes in Tomsk. For this reason, to estimate the tropopause altitude over the lidar site, we use data for vertical temperature profiles from the three nearest to Tomsk meteorological stations launching radiosondes twice a day (at 00:00 and 12:00 UTC)."
**We write**
"To estimate the tropopause altitude over the lidar site, we use vertical temperature profiles from three neighbor meteorological stations launching radiosondes twice a day (at 00:00 and 12:00 UTC)."
[**Page 4, lines 9–10, revised manuscript**]

**Comment 9: page 4, line 7**. What tropopause definition are they using? Please clarify and cite.

**Response 9:** We have added the following sentences to the revised manuscript:

"The lower boundary of the tropopause is determined by the temperature lapse rate of 2 K/km according to the criterion provided by the World Meteorological Organization (WMO, 1957). All temperature profiles and estimated tropopause altitudes that we used in our study are also given in the Supplement."

WMO: World Meteorological Organization: Definition of the tropopause, Bulletin of the World Meteorological Organization, 6, 136–137, 1957.
[**Page 4, lines 14–16, revised manuscript**]

**Comment 10: page 4, line 9**. Move "nearest to Tomsk" to after "stations". Give distances.

**Response 10:** We have corrected Figure 1 and its caption.

**Instead of**
"Three nearest to Tomsk meteorological stations launching radiosondes twice a day."
**We write**
"Three neighbor meteorological stations (with the distances between them and Tomsk) launching radiosondes twice a day."
[**Page 4, line 19, revised manuscript**]

**Comment 11: page 4, line 20**. What is the difference between MPA (maximum plume altitude) and HMPA (presumably "height of maximum plume altitude?")?

**Response 11:** Yes, you are right. MPA is an abbreviation of the maximum plume altitude, whereas $H_{MPA}$ is a designation of the maximum plume altitude (along with $H_0$, $H_1$, $H_2$, $H_{traj.}^{back.}$).

**3.1 Detection of pyroCb smoke plumes in the UTLS**

**Comment 12: page 5, line 20**. 1) Auth do not present any data on the height of the tropopause near the USA endpoints. 2) Also, one cannot come to conclusions regarding MPA from a single plume observation. Maybe the part of the plume that blew over Tomsk wasn't the highest part of the pyroCb plume.

**Response 12:** First, the data on the tropopause altitudes near pyroCb events or places of plume detection in the USA and Canada are now given both in the text and Supplement (see **Response 9**). Second, we agree with this comment and have corrected our conclusion regarding MPA.

**Instead of**
"Based upon the end points of the trajectories (with $H_{traj.}^{back.} \leq 10.5$ km a.g.l.) that are below latitude 45° N (Fig. 2b), the MPA $H_{MPA}$ did not exceed the tropopause at the place of the pyroCb origin."
**We write**
"According to radiosonde data from two close stations located in Green Bay (44.48° N, 88.13° W; USA) and Davenport (41.61° N, 90.58° W; USA), the tropopause was at 15.0–15.9 km on that day. Based upon the end points of the trajectories (with $H_{traj.}^{back.} \leq 10.2$ km) and tropopause altitude, the Jasper Fire smoke came to the Tomsk TR from the UT over the place of the pyroCb plume observation in Iowa."
[**Page 5, lines 27–30, revised manuscript**]

**Comment 13: page 5, line 24**. Why are the quotation marks needed? Consider removing them.

**Response 13:** This comment is no longer relevant due to the changes made in a revised manuscript (see **Response V**).

**Comment 14: page 5, line 27**. Delete "example."

**Response 14:** OK. Done.

**Comment 15: page 7, line 7**. "example" is not necessary. Consider deleting.

**Response 15:** This comment is no longer relevant due to the changes made in a revised manuscript (see the changes after **Response 16**).

**Comment 16: page 7, line 8**. This is vague. Please provide a more detailed characterization of this single observation of the pyroCb plume.

**Response 16:** We agree with this comment and have corrected this part of the manuscript.

**Instead of**
"Based on the behavior of the example trajectories (Fig. 4b) and the tropopause altitudes determined at the three nearest to Tomsk meteorological stations (Fig. 4a), we suppose that the pyroCb plume was spreading in the UT in a given period of time."
**We write**
"Radiosonde data from the Churchill station (58.73° N, 94.08° W; Canada) to the west of the plume registration showed the tropopause altitude of ~11.4 km. According to the Inukjuak station (58.45° N, 78.11° W; Canada) to the east of the plume registration, the tropopause was at ~11.9 km on that day. Thus, we can conclude that the Conibear Lake Fire smoke came to the Tomsk UT/TR from the UT/TR over Hudson Bay."
[**Page 8, lines 1–4, revised manuscript**]

**Comment 17: page 7, line 15**. "only" isn't needed here.  Please consider deleting it, unless auth think it helps their statement.

**Response 17:** We agree. Done.

**Comment 18: page 7, line 16**. Why are quotation ("weak") marks needed?

**Response 18:** As was noted in **Response V**, we have removed from the text all comparative adjectives "weak" and "strong".

**Comment 19: page 8, line 10**. This is confusing because it suggests that there is a first strong layer but none was mentioned. If auth are going to use the terms "weak" and "strong" they should provide a customized definition of them.

**Response 19:** See please **Response 18**.

**Comment 20: page 8, line 19**. There does appear to be a thin cirrus deck over Tomsk according to Aqua MODIS 11μm brightness temperature imagery at 20:15 UTC.  Auth's argument could be bolstered if they showed CALIPSO data from that night. There is an orbit pretty close to Tomsk.

**Response 20:** We agree with this comment. We have analyzed the corresponding CALIPSO data and then have removed the aerosol layer detected on 23 September 2013 in the UTLS over Tomsk from consideration (see also **Response II**).

**Comment 21: page 11 line 4**. Please cite Peterson et al 2018. https://www.nature.com/articles/s41612-018-0039-3.

**Response 21:** We have added this paper to the list of references.

**Comment 22: page 11 line 15**. I don't think observations this distant from the source can be used to connect to an individual pyroCb within that cluster of pyroCbs. It is sufficient just to connect these to the 12 August event in general.

**Response 22:** The HYSPLIT trajectory analysis has attributed all three aerosol layers observed over Tomsk on 26, 29, and 31 August to two pyroCbs detected at (51.8° N, 123.2° W) and (53.1° N, 121.0° W) around 03:30 and 05:30 on 12 August, respectively. However, we do not exclude that the layers could contain aerosol from the other three pyroCbs and note this possibility in the manuscript.

**3.2 Detection of the Bogoslof volcanic plume in 2017**

**Comments 23 and 24** on page 12 are no longer relevant because **Section 3.2 Detection of the Bogoslof volcanic plume in 2017** has been removed from the text (see also **Response II).**

**3.3 PyroCb events in 2004–2012**

**Comment 25: page 14, Table 1**. The plume dates attributable to Guan et al. are the dates of the aerosol index measurement, not the pyroCb.

**Response 25:** We have made some corrections in this section.

**Instead of**
"Several strong pyroCbs, the plumes of which reached UTLS altitudes with $H_{MPA} \geq 12$ km a.s.l. and could potentially be detected in the UTLS over Tomsk, were documented in the Northern Hemisphere between 2004 and 2011 (Table 1). However, no aerosol layers associated with these pyroCb events were observed at the SLS during the period. This was due to unfavorable weather conditions or pyroCb plumes could have diffused or passed by the SLS and, therefore, might not be detected."
**We write**
"Several biomass burning plumes with $H_{MPA} \geq 12$ km, which resulted from pyroCbs and could potentially be detected over Tomsk, were documented in the UTLS of the Northern Hemisphere between 2004 and 2011 (Table 1). However, no aerosol layers associated with these plumes were observed at the SLS during the period. This was due to unfavorable weather conditions (rain, snow, fog, clouds) or pyroCb plumes could have diffused or passed by the SLS and, therefore, might not be detected."
[**Page 12, lines 8–10 and page 13, lines 1–2, revised manuscript**]

**Instead of**
"**Table 1.** List of some documented pyroCbs with $H_{MPA} \geq 12$ km a.s.l. occurred in the Northern Hemisphere, the plumes of which could potentially be detected in the UTLS over Tomsk, in the 2004–2011 period."
**We write**
"**Table 1.** List of biomass burning plumes with $H_{MPA} \geq 12$ km that were documented in the Northern Hemisphere and could potentially be detected in the UTLS over Tomsk in the 2004–2011 period. MPA: maximum plume altitude."
[**Page 13, lines 7–8, revised manuscript**]

**Instead of**
"PyroCb plume location"
**We write**
"Plume location"
[**Page 13, Table 1, revised manuscript**]

**4 Time series of the integrated aerosol backscatter coefficient (2001–2017)**

**Comment 26: page 14, line 10**. What is the rationale for the 10-day averaging? How many data points are typically averaged together? It would be beneficial to have some discussion of the frequency of good aerosol-only profiles should be included. If that information is contained in a prior paper, please mention and cite it.

**Response 26:** Three to four measurements on average are made within a month. The rationale for the use of the 10-day averaging is presented below.

**Instead of**
"To estimate the contribution of the pyroCb events discussed above to aerosol loading of the UTLS over Tomsk, we have analyzed the 2001–2017 time series of the aerosol backscatter coefficient $B_\pi^a$ values, obtained from the SLS observations at $\lambda = 532$ nm and integrated over the 11–30 km altitude range. The upper part of Fig. 12 presents the 10-day average $B_\pi^a$ values with the annual average $B_\pi^a$ ones assigned to 1 July of each year. PyroCb events and volcanic eruptions (Tables 2

and 3), the plumes of which were observed in the UTLS over Tomsk between 2000 and 2017, are indicated by red and black vertical bars, respectively, in the lower part of Fig. 12."

**We write**

"To estimate the contribution of the pyroCb events discussed above to aerosol loading of the UTLS over Tomsk, we have analyzed the 2001–2017 time series of the annual average $B_{\pi,532}^{a}$ values (see Sect. 2). The upper part of Fig. 10 presents both the 10-day and annual average $B_{\pi,532}^{a}$ values obtained from the SLS observations. Due to weather conditions in Tomsk, the observations are often irregular in time and periods without lidar measurements can last up to several months. To obtain a homogeneous time series of $B_{\pi,532}^{a}$ values for the time intervals when measurement data are available, all the data for every 10-day period are averaged. The average values for the periods from days 1 to 10, 11 to 20, and 21 to 30 (31) of a month are assigned to the 5th, 15th, and 25th days of the month, respectively. The same data processing method was used in (Zuev et al., 1998, 2017). The annual average $B_{\pi}^{a}$ values are assigned to 1 July of the corresponding year. PyroCb events and volcanic eruptions (Tables 2 and 3), the plumes of which were observed in the UTLS over Tomsk between 2000 and 2017, are indicated by red and black vertical bars, respectively, in the lower part of Fig. 10."

[**Page 13, lines 10–19, revised manuscript**]

**Comment 27: page 15, line 3**. This sentence is grammatically flawed. It states that 6 pyroCbs "injected smoke" but also "resulted in a negative trend." Perhaps this should be reworded to state that a negative trend was observed in spite of the several pyroCb injections?

**Response 27:** We agree and have reworded this sentence.

**Instead of**

"Namely, only two volcanic eruptions that could perturb the UTLS over Tomsk occurred for a given period of time (Table 3). Six pyroCb events injected smoke into the UTLS in 2013 and 2015–2017 (Table 2) resulted, however, in a negative trend in the annual average $B_{\pi}^{a}$ values."

**We write**

"Only the 2014 Mt. Kelut volcanic eruption could slightly perturb the UTLS over Tomsk in a given period of time (Table 3). Thus, a negative trend in the annual average $B_{\pi,532}^{a}$ values was observed in spite of five pyroCbs that injected smoke into the UTLS in 2013 and 2015–2017 (Table 2)."

[**Page 14, lines 6–9, revised manuscript**]

**Comment 28: page 15, line 6**. It is important to acknowledge, either in this section, or up front, that the single wavelength lidar with no depolarization information content is inadequately constrained for composition assessment. Therefore the conclusions drawn here come with large uncertainty.

**Response 28:** We have noted this moment in the text.

**Instead of**

"PyroCbs generated by wildfires from 2004 to 2011 (including documented ones listed in Table 1) also had to perturb the UTLS over Tomsk, but we could not unambiguously discern the pyroCb plumes against the background of more powerful volcanic plumes observed during this period. Therefore, the positive trend in the period 2004 to 2011 should have been mostly caused by volcanic eruptions (the same conclusion was reached by Zuev et al. (2017), when integrating $\beta_{\pi}^{a}(H,\lambda)$ over the 15–30 km altitude range)."

**We write**

"PyroCbs generated by wildfires from 2004 to 2011 (including documented ones listed in Table 1) also had to perturb the UTLS over Tomsk. But the use of our single-wavelength lidar with no depolarization information makes it impossible to unambiguously discern the pyroCb plumes against the background of more powerful volcanic plumes for the same period. Nevertheless, a comparison of the annual average $B_{\pi,532}^{a}$ values in periods (a) and (c) of volcanic quiescence with those in period (b) of volcanic activity shows that the positive trend in the period 2004–2011 should have been mostly caused by volcanic eruptions. The same conclusion was reached by Zuev et al. (2017) when integrating $\beta_{\pi,532}^{a}(H)$ over the 15–30 km altitude range."

[**Page 14, lines 15–21, revised manuscript**]

**Comment 29: page 15, line 7**. Insert "that".

**Response 29:** OK. Done.

**Comment 30: page 16 line 2.** The black sloping lines are not described in this caption.

**Response 30:** We have added the required description to the figure caption.

**The following sentence was added to Fig. 10 capture**

"The black sloping lines show the trends in the annual average $B_{\pi,532}^{a}$ values for the 2001–2004, 2004–2011, and 2011–2017 periods."

[**Page 15, line 6, revised manuscript**]

**6 Data availability**

**Comment 31: page 18, line 24**. Are the lidar data available?

**Response 31:** We have added both the 10-day and annual average $B_{\pi,532}^{a}$ values obtained from the SLS observations to supplementary materials.

Sincerely,
Authors

---

## Author Comment (AC3) · 28 Feb 2019

**Manuscript Number:** acp-2016-1153
**Manuscript Type:** Research article

**Title:** Lidar observations of pyrocumulonimbus smoke plumes in the UTLS over Tomsk (Western Siberia, Russia) from 2000 to 2017

**List of corrections**

**General comments**

**1.** Figures 6a, 6b, 11a, and 11b were removed from the manuscript; the other figures were substituted by new ones. Captions of all figures were corrected. The figures were renumbered starting with Fig. 7, i.e.:

Fig. 7 → Fig. 6,
Fig. 8 → Fig. 7,
Fig. 9 → Fig. 8,
Fig. 10 → Fig. 9,

Fig. 12 → Fig. 10

**2.** The parts of the text concerning the aerosol layer observed over Tomsk **on 23 September 2013 (page 8)** and **Section 3.2 Detection of the Bogoslof volcanic plume in 2017 (pages 12–13)** were removed from the revised manuscript, i.e.:

**"**~~The second "strong" layer was observed over Tomsk at altitudes between 11.2 and 12.8 km with the maximum $R(H) =$ 11.4 at $H = 11.8$ km a.s.l. on 23 September 2013 (Fig. 6a). A trajectory analysis showed that the layer can be assigned to a pyroCb event observed in British Columbia (~54° N, ~126° W; Canada) using the GOES-15 visible, shortwave IR, and longwave IR imageries between 23:30 UTC on 15 September and 02:30 UTC on 16 September (http://pyrocb.ssec.wisc.edu/archives/272). Three example HYSPLIT air mass backward trajectories that started from altitudes of ~12.15 km a.s.l. over Tomsk at 17:30 UTC on 23 September and then passed close to the place of the pyroCb origin at altitudes $H_{\text{traj.}}^{\text{back.}} \approx 10.7$–11.7 km a.g.l. on 16 September are shown in Fig. 6b. Despite the high value of the scattering ratio $R(H)$, which is representative of cirrus clouds, the tropopause altitudes determined at the nearest to Tomsk meteorological stations show that the aerosol layer maximum was definitely in the LS (Fig. 6a). This allows us to conclude that the layer was a stratospheric one and could not be a cirrus cloud~~.**"**
**and Section 3.2 → Section 3.1.**

**3.** The notations $R(H, \lambda)$, $\beta_\pi^{\text{m}}(H,\lambda)$, $\beta_\pi^{\text{a}}(H,\lambda)$, and $B_\pi^{\text{a}}(\lambda)$ were substituted by the $R_{532}(H)$, $\beta_{\pi,532}^{\text{m}}(H)$, $\beta_{\pi,532}^{\text{a}}(H)$, and $B_{\pi,532}^{\text{a}}$ ones, respectively, throughout the revised manuscript (including Eqs. (1) and (2)). The shorter notation $R(H)$ is also used in **Section 3.1** and **Appendix A**.

**4.** All the HYSPLIT air mass backward trajectories were recalculated "above mean sea level" (AMSL). Therefore, all altitudes are now given AMSL, while all abbreviations **a.s.l.** and **a.g.l.** were removed from the revised manuscript.

**5.** All dates and times are given in **UTC** and the abbreviation **UTC** was removed from the revised manuscript with the exception of figure captions.

**6.** The adjectives "**weak**" and "**strong**" applied for aerosol layers were removed from the revised manuscript.

**Page 1**

**Instead of**
"Using the HYSPLIT trajectory analysis, we have reliably assigned ten aerosol layers to nine out of more than 100 documented pyroCb events, the aftereffects of which could potentially be detected in the UTLS over Tomsk. All of the nine pyroCb events occurred in the USA and Canada: one event per year was in 2000, 2002, 2003, 2015, and 2016, whereas two events per year were in 2013 and 2017. No plumes from pyroCbs originating in the boreal zone of Siberia and the Far East (to the east of Tomsk) were observed in the UTLS over Tomsk between 2000 and 2017. We conclude that the lifetimes of pyroCb plumes to be detected in the UTLS using ground-based lidars are less than about a month, i.e. plumes from pyroCbs generated by wildfires to the east of Tomsk can significantly diffuse before reaching the Tomsk lidar station by the westerly zonal transport of air masses. A comparative analysis of the contributions from pyroCb events and volcanic eruptions with VEI ≥ 3 to aerosol loading of the UTLS over Tomsk has also been made. Finally, an aerosol plume from the Aleutian volcano Bogoslof erupted with VEI = 3 on 28 May 2017 was detected at altitudes between 10.8 and 13.5 km over Tomsk on 16 June 2017."

**we wrote**
"Using the HYSPLIT trajectory analysis, we have reliably assigned nine aerosol layers to eight out of more than 100 documented pyroCb events, the aftereffects of which could potentially be detected in the UTLS over Tomsk. All the eight pyroCb events occurred in the USA and Canada: one event per year was in 2000, 2002, 2003, 2013, 2015, and 2016, whereas two events were in 2017. No plumes from pyroCbs originating in the boreal zone of Siberia and the Far East (to the east of Tomsk) were observed in the UTLS over Tomsk between 2000 and 2017. We conclude that the time durations for pyroCb plumes to be detected in the UTLS using ground-based lidars are less than about a month, i.e. plumes from pyroCbs generated by wildfires to the east of Tomsk can significantly diffuse before reaching the Tomsk lidar station by the westerly zonal transport of air masses. A comparative analysis of the contributions from pyroCb events and volcanic eruptions with VEI ≥ 3 to aerosol loading of the UTLS over Tomsk showed the following. Plumes from two or more pyroCbs that have occurred in North America in a single year are able to markedly increase the aerosol loading compared to the previous year. The annual average value of the integrated aerosol backscatter coefficient $B_{\pi,532}^{\mathrm{a}}$ increased by 14.8% in 2017 compared to that in 2016 due to multiple pyroCbs occurred in British Columbia (Canada) in August 2017. The aftereffects of pyroCb events are comparable to those of volcanic eruptions with VEI ≤ 3, but even multiple pyroCbs can hardly compete with volcanic eruptions with VEI = 4."
[**Page 1, lines 15–28, revised manuscript**]

**Page 3**

**Instead of**
"To consider the effect of only volcanic eruptions on stratospheric aerosol loading and definitely exclude from consideration any aerosol perturbations in the upper troposphere (UT) (such as cirrus clouds) and tropopause region, we analyzed the results of lidar measurements at altitudes higher than 13–15 km. It is clear that this altitude limitation could lead to the loss of information on aerosol events like pyroCb plumes in the UTLS over Tomsk.
The possibility to observe stratospheric smoke plumes in Tomsk from massive forest fires occurred in North America was noted in Zuev et al. (2017). In this paper, we analyze all aerosol perturbations in the 11–30 km altitude region over Tomsk that could be caused by massive wildfires in North America and North-East Asia from 2000 to 2017."
**we wrote**
"To consider the effect of only volcanic eruptions on stratospheric aerosol loading and definitely exclude from consideration any aerosol perturbations in the upper troposphere (UT) (such as cirrus clouds) and tropopause region (TR), we analyzed the results of lidar measurements at altitudes higher than 13–15 km. It is clear that this altitude limitation could lead to the loss of information on aerosol events like pyroCb plumes in the UTLS over Tomsk. The possibility to observe stratospheric smoke plumes in Tomsk from massive forest fires occurred in North America was noted in Zuev et al. (2017). In this paper, we analyze aerosol perturbations in the 11–30 km altitude region over Tomsk that could be caused by massive wildfires in North America and North-East Asia from 2000 to 2017."
[**Page 3, lines 4–11, revised manuscript**]

**Instead of**
"Here $\beta_{\pi}^{\mathrm{m}}(H,\lambda)$ and $\beta_{\pi}^{\mathrm{a}}(H,\lambda)$ are the altitude- and wavelength-dependent molecular…"
**we wrote**
"Here $\beta_{\pi,532}^{\mathrm{m}}(H)$ and $\beta_{\pi,532}^{\mathrm{a}}(H)$ are the altitude-dependent molecular…"
[**Page 3, line 23, revised manuscript**]

**Page 4**

**Instead of**

[revised manuscript text omitted]

---

## Author Comment (AC4) · 28 Feb 2019

The comment was uploaded in the form of a supplement:
https://www.atmos-chem-phys-discuss.net/acp-2018-1153/acp-2018-1153-AC4-supplement.zip